# Identification of novel genes responsible for a pollen killer present in local natural populations of *Arabidopsis thaliana*

Anthony Ricou[1], Matthieu Simon[1], Rémi Duflos[2], Marianne Azzopardi[1], Fabrice Roux[2], Françoise Budar[1], Christine Camilleri[1]*

1 Université Paris-Saclay, INRAE, AgroParisTech, Institute Jean-Pierre Bourgin for Plant Sciences (IJPB), Versailles, France, 2 LIPME, Université de Toulouse, INRAE, CNRS, Castanet-Tolosan, France

* christine.camilleri@inrae.fr

**Data Availability Statement:** All relevant data are within the manuscript and its Supporting Information files.

## Abstract

Gamete killers are genetic loci that distort segregation in the progeny of hybrids because the killer allele promotes the elimination of the gametes that carry the sensitive allele. They are widely distributed in eukaryotes and are important for understanding genome evolution and speciation. We had previously identified a pollen killer in hybrids between two distant natural accessions of *Arabidopsis thaliana*. This pollen killer involves three genetically linked genes, and we previously reported the identification of the gene encoding the antidote that protects pollen grains from the killer activity. In this study, we identified the two other genes of the pollen killer by using CRISPR-Cas9 induced mutants. These two genes are necessary for the killer activity that we demonstrated to be specific to pollen. The cellular localization of the pollen killer encoded proteins suggests that the pollen killer activity involves the mitochondria. Sequence analyses reveal predicted domains from the same families in the killer proteins. In addition, the C-terminal half of one of the killer proteins is identical to the antidote, and one amino acid, crucial for the antidote activity, is also essential for the killer function. Investigating more than 700 worldwide accessions of *A. thaliana*, we confirmed that the locus is subject to important structural rearrangements and copy number variation. By exploiting available *de novo* genomic sequences, we propose a scenario for the emergence of this pollen killer in *A. thaliana*. Furthermore, we report the co-occurrence and behavior of killer and sensitive genotypes in several local populations, a prerequisite for studying gamete killer evolution in the wild. This highlights the potential of the Arabidopsis model not only for functional studies of gamete killers but also for investigating their evolutionary trajectories at complementary geographical scales.

## Author summary

Certain genetic elements are qualified as selfish because they favor their transmission to the progeny during reproduction to the detriment of gametes that do not carry them. These elements are widespread in fungi as well as in plants or in animals, and they are made up of two or even three components, which are specific to each species. Therefore,

**Funding:** R.D. was funded by a grant from the French Ministry of National Education and Research. The funders had no role in study design, data collection and analysis, decision to publish, or preparation of the manuscript.

**Competing interests:** The authors have declared that no competing interests exist.

they must be studied on a case-by-case basis. Moreover, understanding how they appear and propagate in local populations remains a major issue in evolutionary biology. Here we have characterized, in the model plant Arabidopsis, the three genes involved in such an element, called a pollen killer. This pollen killer targets the mitochondria to cause the death of pollen grains that do not carry it. We investigated the three genes in several hundred genotypes collected worldwide, giving us a global view of their diversity at the species level. We also found that some French local populations contain both sensitive and killer plants, which constitutes an invaluable resource for studying the evolution of a pollen killer in the wild.

## Introduction

The study of non-Mendelian inheritance has been instrumental to seminal genetic discoveries, from chromosomes to transposable elements [1,2]. Much can still be expected from studies on genetic loci that subvert Mendel's inheritance laws, causing segregation distortion in hybrid progenies, particularly on chromosome evolution and male gamete development [3,4]. Such loci have been reported in many eukaryotes, from fungi to plants and animals [5,6]. Loci that distort segregation may act at meiosis, and are named meiotic drivers, or by gamete elimination, and are named gamete killers [7]. In plants, gamete killers have been documented mainly in crops because they complicate the use of genetic resources for breeding: most of them concern rice inter-specific or -subspecific crosses (reviewed in [8]), but also vegetable and cereals [9–15]. Yet, a few have been reported in wild species [16,17], including several pollen killers in *Arabidopsis thaliana* [18]. Several distorters have been studied at the molecular level in fungi (reviewed in [7]), in animals [19–21], and in vascular plants, where gamete killers are called pollen killers when they affect male gametes. Causal genes for plant segregation distorters have been identified mostly in rice [22–28], but also in maize [29,30] and yellow monkeyflower [31]. However, very few mechanisms were deciphered at the molecular level [27,30]. Gamete killers can be divided into poison-antidote systems, where the killer allele produces both a poison and its antidote, and killer-target systems, where a target is encoded by the eliminated (sensitive) allele [32]. Studies of gamete killer loci have revealed several common characteristics [33]: (i) except in certain fungi, poison-antidote systems are encoded by at least two genes [7,34]; (ii) the genes involved are genetically linked, which avoids the generation of 'suicidal' haplotypes encoding the poison but not the antidote [35]; (iii) these genes are mainly species-specific; (iv) gamete killer loci are usually subject to important structural rearrangements and copy number variation [32,33,36,37].

Gamete killers are thought to contribute to genome evolution and induce genomic conflicts [38], which can lead to genetic isolation and speciation [5,39,40] and drive sex chromosome evolution [41]. Understanding how these systems arise and propagate in a population remains a major question in evolutionary biology [42]. Indeed, cheating alleles that promote their transmission during gametogenesis are predicted to spread in populations [33]. Studies in natural populations are thus necessary to test predictions based on population genetic models. In addition, gene drive systems are currently adapted or developed with the aims to control weeds in the field or invasive plant species in natural populations [43], as for animal disease vectors, such as mosquitoes [44,45]. The use of such technologies in natural ecosystems and agroecosystems will need a clear governance framework, which has to rely on the knowledge of the population dynamics and evolutionary characteristics of gene drive in natural conditions [43,46]. This can be provided only by studying natural populations where distorters with

identified causal genes are active. A limited number of such studies have been conducted, essentially in populations of *Mimulus guttatus* [47]. *A. thaliana*, a species for which numerous molecular tools and an increasing number of sampled natural polymorphic populations are available [48,49], represents a real opportunity for such studies.

In a first study [18], we identified in *A. thaliana* a pollen killer locus located at the bottom of chromosome 3. This pollen killer, named PK3, was first observed in the progeny of hybrids between two geographically distant natural accessions, Mr-0 (originating from Sicily) and Shahdara (Sha, from Tajikistan), where it induces a deficit of homozygous Sha plants at the locus. Sha thus carries the sensitive allele, whereas Mr-0 has the killer one. We then showed that this pollen killer works according to a poison-antidote model and characterized the anti-dote gene [50]. Some non-killer accessions, such as Col-0, nevertheless possess the antidote function and are resistant, *i.e.* no bias is observed in their F2 progenies either in crosses with Sha or in crosses with Mr-0. We identified killer, sensitive, and resistant behaviors amongst accessions sampled from a worldwide collection. Genomic sequencing of a subset of these accessions showed that the PK3 locus was hypervariable, with important structural variations, particularly in killer genotypes (Fig 1). Genetic analyses in the cross Mr-0 x Sha revealed that a central region of the locus, named PK3B, contains the antidote gene and is flanked by two regions, named PK3A and PK3C, each carrying an element necessary for the killer activity [50]. The gene encoding the antidote was identified in Col-0 as *AT3G62530*, and named *APOK3*. This gene originated and has evolved in the species *A. thaliana*; it is present in two or three copies in the killer genotypes. *APOK3* encodes a chimeric mitochondrial protein whose sequence differs between sensitive and resistant accessions, and three amino acid changes linked to its antidote capacity were identified [50].

In the present work, we characterize the two killer genes of the PK3 locus, investigate the diversity of the three genes (antidote and killer genes) on a worldwide scale, and explore the occurrence and behavior of this pollen killer within and between several polymorphic French local populations.

## Results

### Identification of PK3 killer components in PK3A and PK3C

The identification of the killer genes was complicated by the need of two elements for the killer activity, one located in the PK3A interval and the other in the PK3C interval [50]. We therefore

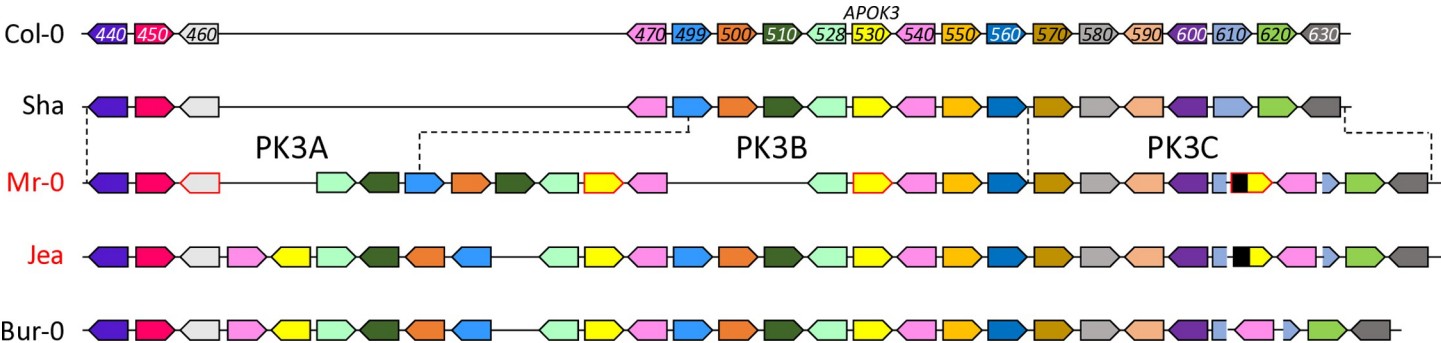

**Fig 1. Schematic representation of the PK3 locus in diverse *A. thaliana* accessions (from [50]).** Names of killer and non-killer accessions are written in red and in black, respectively. Dashed lines indicate the limits in Sha and Mr-0 of the genetically determined PK3A, PK3B and PK3C intervals. In the killer alleles, PK3A and PK3C each contain at least one element required for the killer activity. In PK3B, *APOK3* is the antidote gene, whose sequence differs between sensitive and resistant accessions. PK3 alleles are drawn to highlight synteny between protein coding genes, they are not to scale and TEs are not represented. Plain colored arrows represent coding genes with their orientations, each color refers to orthologs and paralogs of a same gene. Sequences homologous to *APOK3* are in yellow. Genes studied in this work are framed in red on the Mr-0 allele. Gene labels correspond to the last three digits of *AT3G62xxx* gene identifiers in Col-0 (TAIR10).

focused, in each interval, on features that were shared by killer accessions, but not necessarily absent in non-killer ones. In the PK3A interval (Fig 1), we identified the gene homologous to *AT3G62460* in Mr-0 as a good killer candidate. It is polymorphic between Sha and Mr-0, and the four other killer accessions previously sequenced—Etna-2, Cant-1, Ct-1 and Jea [50]—had the same polymorphisms as Mr-0 with respect to Sha. The Sha and Mr-0 protein sequences differed by 14 amino acid changes (Table 1). In addition, the sequence of the gene in Sha presents a 11-nucleotide insertion located 60 bp upstream of the ATG starting position and a large insertion of 1308 bp in the intron compared to Mr-0. To test whether this gene is necessary for the killer activity, we used a CRISPR-Cas9 strategy to obtain loss of function mutants. Mutagenesis was performed in the genetic background ShaL3$^{M}$, Mr-0 for the PK3 locus and Sha for the rest of the nuclear genome, previously used to genetically decipher PK3 [50]. Two independent knock-out (KO) mutants, named *460#1* and *460#2* (Fig 2A), were analyzed. Homozygous mutants *460#1* and *460#2* both had normal phenotypes indistinguishable from wild-type plants and were fully fertile, with no dead pollen visible after Alexander staining of anthers (S1 Fig). Plants heterozygous for each mutation were crossed with Sha, and we analyzed the progeny of F1 plants with the mutated Mr-0 allele, plus one plant without mutation as a positive control for the killer activity. Unlike the control, the hybrids with mutations showed no dead pollen (S1 Fig), and no bias was observed against the Sha allele at the PK3 locus in their F2 progenies (Fig 2C and S1 Table). The killer activity of the Mr-0 allele was thus abolished in these mutants, showing that *AT3G62460* is a killer gene, necessary for the PK3 activity. We therefore named it *KPOK3A*, for **K**ILLER OF **PO**LLEN **K**ILLER IN PK**3A**.

In the PK3C interval, a striking feature shared by Mr-0 and the four other killer accessions previously sequenced [50] was that the second intron of *AT3G62610* contains a very large insertion of over 10 kb (Fig 1 and S1 File). This insertion includes a sequence homologous to the antidote gene *APOK3*, which was named *APOK3-like* [50]. The whole locus structure of the non-killer accession Bur-0 is very similar to that of the killer accessions, except that the sequence *APOK3-like*, present in all the killers, is missing in Bur-0 (Fig 1 and S1 File). *APOK3-like* was thus a suitable killer candidate in the PK3C interval. The EuGene software [51] predicted that *APOK3-like* consists of a novel 600-bp sequence fused 5' to the last 537 bp of *APOK3* (Fig 2B). We confirmed this structure by 5'-RACE and RT-PCR experiments (S2 Fig and S1 File). As for *KPOK3A*, we performed a CRISPR-Cas9 mutagenesis of *APOK3-like* in the genetic background ShaL3$^{M}$ and analyzed two independent KO mutants, obtained by using two different guide-RNAs (Fig 2B). The homozygous mutants *apok3-like#1* and *apok3-like#2* showed no phenotypic alterations compared to the wild-type, were perfectly fertile and showed no pollen mortality (S1 Fig). In the same way as for *KPOK3A*, heterozygous mutants were crossed with Sha, and we analyzed F1 plants with the mutations, plus one control without mutation. We observed no dead pollen in the mutated F1s (S1 Fig) and their F2 progenies showed no bias against the Sha allele at the PK3 locus, while the control did (Fig 2C and S2 Table). Therefore, inactivating the gene *APOK3-like* suppresses the killer activity: this gene is a mandatory killer element of the PK3C interval, and we renamed it *KPOK3C*.

According to the poison-antidote model, the killer genes are expressed in diploid cells to produce the poison, which subsists in all the gametes. RT-PCR assays were performed on

**Table 1. AT3G62460 amino acid changes in Mr-0 and Sha compared to Col-0.**

| amino acid position | 6 | 9 | 17 | 43 | 46 | 51 | 53 | 55 | 60 | 65 | 84 | 89 | 107 | 112 | 139 | 166 |
|---|---|---|---|---|---|---|---|---|---|---|---|---|---|---|---|---|
| Col-0 | C | C | I | - | D | H | R | E | S | V | N | K | Q | K | V | G |
| Mr-0 | F | C | L | - | D | H | R | E | S | V | N | K | Q | K | I | D |
| Sha | C | F | L | N | Y | D | P | K | G | L | K | E | R | N | V | D |

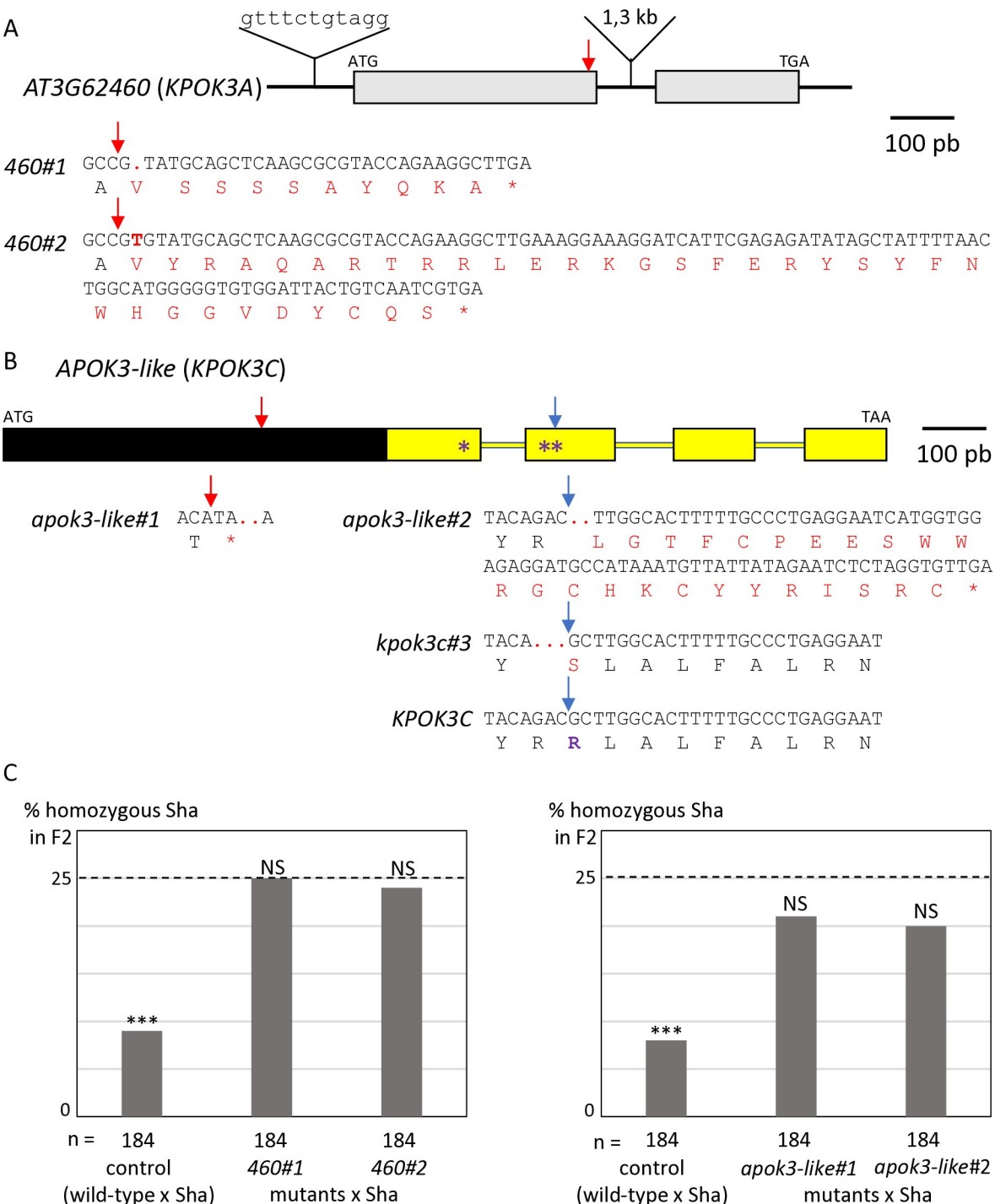

**Fig 2. Identification of the PK3 killer genes. A & B.** Structure of the candidate killer genes. Boxes represent coding sequences, lines represent introns and UTRs. Red and blue arrows indicate target sites of CrispR-Cas9 mutagenesis. The coding sequences of the mutants are given below the genes, with the predicted protein sequences. Modified residues are in red, deletions are indicated by dots, inserted nucleotides are in bold. **A**. Structure of *KPOK3A* (*AT3G62460*) Mr-0 gene. Indels that differentiate the Sha *KPOK3A* allele are represented above the gene. The mutants *460#1* and *460#2* have respectively a G deletion and a T insertion at 353 bp from the ATG, leading to proteins modified from amino acid 118 and stopped prematurely after 127 (for *460#1*) and 151 (for *460#2*) amino acids. **B**. Structure of *APOK3-like* (*KPOK3C*) Mr-0 gene. The part of the gene identical to the antidote gene *APOK3* is in yellow. Purple stars indicate the positions of the S, D, and R residues specific for the functional form of the antidote. The mutant *apok3-like#1*, mutated in the region specific for *KPOK3C*, has a 2-bp (CC) deletion at 408 bp from the ATG, leading to a protein stopped after 135 amino acids. The mutants *apok3-like#2* and *kpok3c#3* are mutated in the region of *KPOK3C* that is identical to *APOK3*. *apok3-like#2* has a 2-bp (GC) deletion at 858 bp from the ATG,

leading to a protein modified from amino acid 263 and stopped after 286 amino acids. *kpok3c*#3 has a 3-bp deletion that does not result in a frameshift. The Mr-0 wild-type *KPOK3C* is shown at the bottom, the R263 residue, corresponding to the R105 residue of the functional form of the antidote, is in purple and bold. **C**. Percentages of plants homozygous for the Sha allele at the PK3 locus in selfed progenies of crosses between Sha and KO mutants of *AT3G62460* (left panel) or of *APOK3-like* (*KPOK3C*) (right panel). Control: wild-type F1 sibling. The percentage expected in the absence of segregation distortion (25%) is indicated by a dotted line on each panel. n: number of F2 plants genotyped for each cross. *** p < 0.001; NS, not significant. Complete data are shown in S1 and S2 Tables.

RNAs isolated from leaves and floral buds of different genotypes. *KPOK3A* was expressed at a comparable level between Mr-0 and genotypes that have the same allelic form, such as the recombinant Rec5 (Mr-0 at PK3A and PK3B and Sha for the rest of the nuclear genome, including PK3C (Fig 3A)) or Bur-0 (Figs 3B and S3A). On the contrary, it was expressed at a much lower level in both Sha and Col-0, whose *KPOK3A* alleles are different from that of Mr-0 (Figs 3B and S3A). Both Sha and Col-0 allelic forms of *KPOK3A* possess the 11 bp located 60 bp upstream of the ATG starting position that is absent in Mr-0, in other killers, and in Bur-0 (Fig 3D). The different expression levels between Sha and Col-0 on the one hand and Mr-0 and Bur-0 on the other could thus be related to this 11-bp sequence, which is located next to a putative TATA box (Fig 3D). RT-PCR assays showed that *KPOK3C* is expressed, although not very strongly, in floral buds (Fig 3C) and leaves (S3B Fig) of Mr-0 and three other killer accessions.

To test if *KPOK3A* was the only gene required for the killer activity in the PK3A interval, we introduced the Mr-0 genomic sequence of *KPOK3A* into the recombinant hybrid background Rec4 x Sha, heterozygous Sha/Mr-0 only at PK3B and PK3C and Sha for the rest of the nuclear genome, including PK3A (Fig 3A). Because the *KPOK3A* Mr-0 allele is not present in Rec4 x Sha, the pollen killer is not active in this genotype [50]. Anthers of plants Rec4 x Sha transformed with *KPOK3A* showed dead pollen after Alexander staining (Fig 3E). In addition, while no bias was observed in the progeny of an untransformed sibling, a segregation bias against the Sha allele was observed at the PK3 locus in the progenies of three independent transformants (Fig 3F and S3 Table). *KPOK3A* was thus sufficient to complement a non-killer allele of the PK3A interval, it is the only killer element necessary in this interval.

In the same way, to test whether *KPOK3C* is the only required killer element in the PK3C interval, its genomic sequence was introduced into the recombinant genotype Rec5 x Sha (Fig 3A), where the pollen killer is not active because Rec5 has no *KPOK3C* [50]. The progenies of six independent heterozygous transformants were analyzed, but none of them showed any bias at the PK3 locus (Table 2). At this stage, we could not exclude that another genetic element located in the PK3C interval is also required for the killer activity. We therefore crossed a T2 plant that carried this construct and was Sha at the PK3 locus with the mutant *apok3-like*#1 (KO mutant in the gene *KPOK3C*, Fig 2B), and did not observe any segregation bias in the progenies of three hybrid plants (Table 2). We concluded that this construct was unable to replace the endogenous *KPOK3C*. We hypothesized that the endogenous promoter sequence used (650 bp upstream of the *KPOK3C* start codon) might not be enough to ensure a proper expression of the transgene. Surprisingly, RT-PCR assays showed that *KPOK3C* was nevertheless expressed in buds of the transformants, and even at a higher level than in the control genotypes Mr-0 and ShaL3$^H$ (Fig 3G). We therefore suspected that the expression was too weak in the meiocytes themselves. We then put the gene under the control of the pRPS5A promoter, described as active in meiocytes [52], and introduced it in both Rec5 x Sha and *apok3-like*#1 x Sha genotypes. We analyzed four and two independent heterozygous transformants respectively, and were unable to conclude to an effective complementation, although a slight bias was observed for one transformed mutant (Table 2). We think that the pRPS5A promoter is not

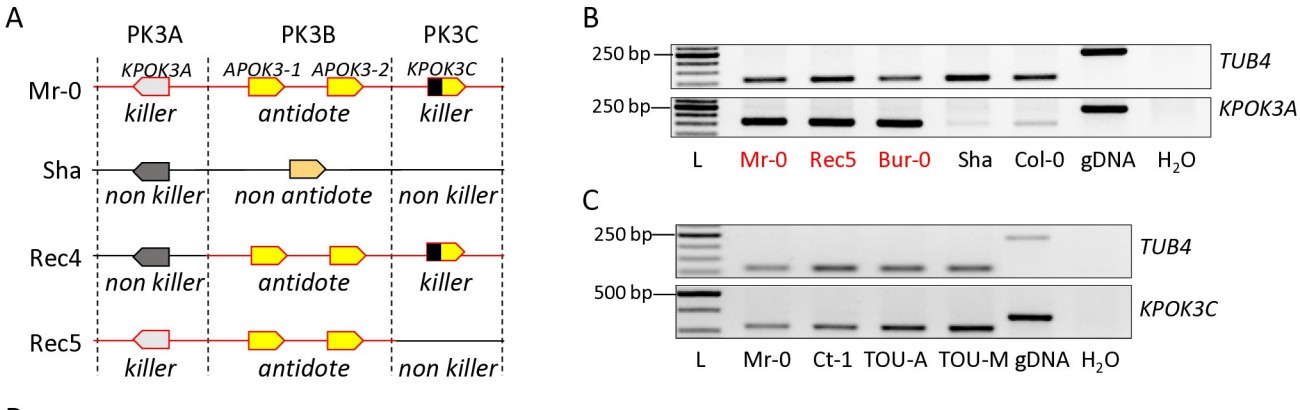

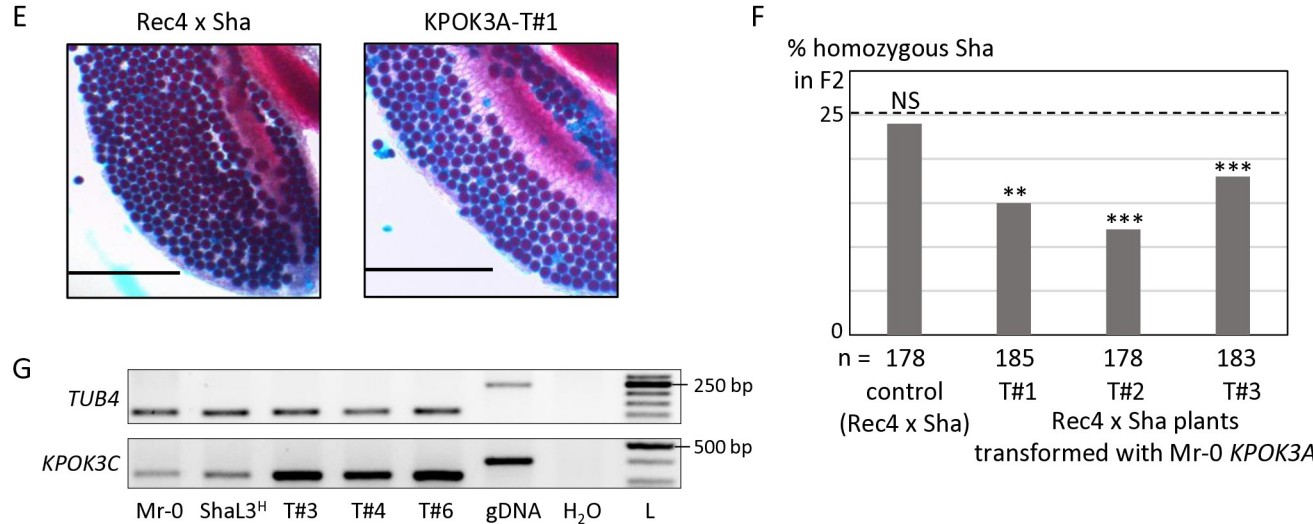

**Fig 3. Expression of *KPOK3A* and *KPOK3C* and complementation of incomplete killer alleles. A.** Simplified representation of Sha, Mr-0 and the recombinant alleles Rec4 and Rec5. Mr-0 alleles are drawn in red and Sha alleles in black. In contrast with the heterozygote Mr-0/Sha, there is no bias in the selfed progenies of the heterozygotes Rec4/Sha or Rec5/Sha, because Rec4 and Rec5 lack the Mr-0 allele of *KPOK3A* and *KPOK3C*, respectively. **B.** Expression of *KPOK3A*. *KPOK3A* PCR amplification of cDNA from buds of plants with different allelic forms of the gene (Mr-0 like, in red, or not, in black). cDNAs were synthetized from 0.2 µg of total RNA; 30 PCR cycles. The primers used were defined on sequences that were invariant between the genotypes tested. *TUB4* (*AT5G44340*) was used as control. gDNA: Mr-0 genomic DNA. L: GeneRuler 50 bp DNA Ladder (Thermo Scientific). **C.** Expression of *KPOK3C*. *KPOK3C* PCR amplification of cDNA from buds of different killer accessions. cDNAs were synthetized from 1 µg of total RNA; 28 PCR cycles for *TUB4* and 32 PCR cycles for *KPOK3C*. TOU-A: TOU-A1-111; TOU-M: TOU-M1-3. **D.** Sequences of the 90 bp upstream of the ATG of *KPOK3A* in Sha and Col-0, and in Mr-0 and Bur-0. The putative TATA box is written in red. **E.** Pollen viability (Alexander coloration of anthers, viable pollen is colored in red and dead pollen appears blue) of plants Rec4xSha (a) and Rec4xSha transformed with the *KPOK3A* Mr-0 allele (b). Scale bars: 200µM. **F.** Percentages of plants homozygous for the Sha allele at the PK3 locus in selfed progenies of Rec4 x Sha plants transformed with the *KPOK3A* Mr-0 allele. The percentage expected in the absence of segregation distortion (25%) is indicated by a dotted line. n: number of F2 plants genotyped for each cross. *** p < 0.001; ** p < 0.01; NS, not significant. Complete data are shown in S3 Table. **G.** *KPOK3C* PCR amplification of cDNA from buds (as in **C**) of three independent T1 (Rec5 x Sha) plants transformed with *KPOK3C* and showing no bias at the PK3 locus in their progenies (T#3, T#4 and T#6: (Rec5 x Sha)_KPOK3C#3, #4 and #6 respectively, Table 2). ShaL3$^H$: heterozygous Sha/Mr-0 for the PK3 locus and Sha for the rest of the nuclear genome.

suitable for mimicking *KPOK3C* endogenous expression, and we still cannot rule out the presence of an additional mandatory killer element in the PK3C interval.

**Table 2. Segregations at the PK3 locus in selfed progenies of recombinants and mutants transformed with *KPOK3C* under the control of its own promoter or of the pRPS5A promoter.**

| Hybrid | Number of F2 plants | | | | $p\ \chi 2$ (Mr:Hz:Sha = 1:2:1) | $f$ Sha [a] |
|---|---|---|---|---|---|---|
| | Total | Mr-0 | Hz | Sha | | |
| Rec5 x Sha (control) | 184 | 51 | 89 | 44 | 0.69[NS] | 24% |
| (Rec5 x Sha)_*KPOK3C*#1 | 183 | 43 | 99 | 41 | 0.53[NS] | 22% |
| (Rec5 x Sha)_*KPOK3C*#2 | 181 | 42 | 85 | 54 | 0.32[NS] | 30% |
| (Rec5 x Sha)_*KPOK3C*#3 | 181 | 47 | 88 | 46 | 0.93[NS] | 25% |
| (Rec5 x Sha)_*KPOK3C*#4 | 181 | 50 | 82 | 49 | 0.45[NS] | 27% |
| (Rec5 x Sha)_*KPOK3C*#5 | 180 | 42 | 95 | 43 | 0.75[NS] | 24% |
| (Rec5 x Sha)_*KPOK3C*#6 | 181 | 43 | 98 | 40 | 0.51[NS] | 22% |
| (apok3-like#1 x Sha)_*KPOK3C*#1 | 182 | 48 | 92 | 42 | 0.81[NS] | 23% |
| (apok3-like#1 x Sha)_*KPOK3C*#2 | 168 | 48 | 74 | 46 | 0.30[NS] | 27% |
| (apok3-like#1 x Sha)_*KPOK3C*#3 | 181 | 53 | 92 | 46 | 0.51[NS] | 25% |
| (Rec5 x Sha)_pRPS5-*KPOK3C*#1 | 368 | 85 | 202 | 85 | 0.25[NS] | 23% |
| (Rec5 x Sha)_pRPS5-*KPOK3C*#2 | 372 | 84 | 196 | 92 | 0.49[NS] | 25% |
| (Rec5 x Sha)_pRPS5-*KPOK3C*#3 | 372 | 82 | 187 | 103 | 0.30[NS] | 28% |
| (Rec5 x Sha)_pRPS5-*KPOK3C*#4 | 372 | 100 | 186 | 86 | 0.59[NS] | 23% |
| (*apok3-like*#1 x Sha)_pRPS5-*KPOK3C*#1 | 368 | 104 | 178 | 86 | 0.34[NS] | 23% |
| (*apok3-like*#1 x Sha)_pRPS5-*KPOK3C*#2 | 372 | 114 | 181 | 77 | 0.02[*] | 21% |

[a] Percentage of plants homozygous for the Sha allele

Plants were genotyped with the marker msat3.23007 (S16 Table)

[*] $p < 0.05$; NS, not significant

## PK3 proteins have similar domains and are associated with mitochondria

The protein sequences most similar to KPOK3A retrieved from public databases through BLAST P search were found in *Arabidopsis suecica* (NYN limkain-b1-type, 91% identity) and *Arabidopsis arenosa* (unnamed protein, 90% identity). In *A. thaliana*, KPOK3A is annotated as a putative endonuclease or glycosyl hydrolase (Araport11, https://www.arabidopsis.org)and it contains a predicted NYN domain (IPR021139 amino acids 58 to 178), and a LabA-like PIN domain of limkain b1 (cd10910, amino acids 58 to 168) (Fig 4A). Although KPOK3A and KPOK3C have no sequence homology, domains of the same families were predicted in both proteins: KPOK3C also contains, in its N-terminus moiety, a predicted NYN domain (IPR021139, amino acids 10 to 87), included in a LabA-like PIN domain of limkain b1 (cd10910, amino acids 2 to 105). As for APOK3, we did not find any ortholog for KPOK3C in the sequences available in public databases. KPOK3C is strictly identical to APOK3 in its last 179 amino acids, which include three HEAT motifs and the three residues which were associated with the antidote activity of APOK3 [50] (Fig 4A). In addition, KPOK3A is identical to the N-terminus of APOK3 for the first 44 amino acids, which include the mitochondria-targeting peptide (Fig 4A). We thus expected KPOK3A to be located in the mitochondria, as previously shown for APOK3 [50]. Indeed, the colocalization of KPOK3A and APOK3 was confirmed using translational fusions with fluorescent proteins (Fig 4B). Unlike APOK3 and KPOK3A, no N-terminal mitochondria-targeting peptide was predicted in KPOK3C by TargetP (likelihood = 0; https://services.healthtech.dtu.dk/services/TargetP-2.0/).However,the recently developed software MULocDeep (https://www.mu-loc.org/ [53]) predicted KPOK3C to be located in mitochondria (p = 0.47) with a possible location in the nucleus (p = 0.24). To experimentally determine KPOK3C localization, we fused its full-length genomic sequence

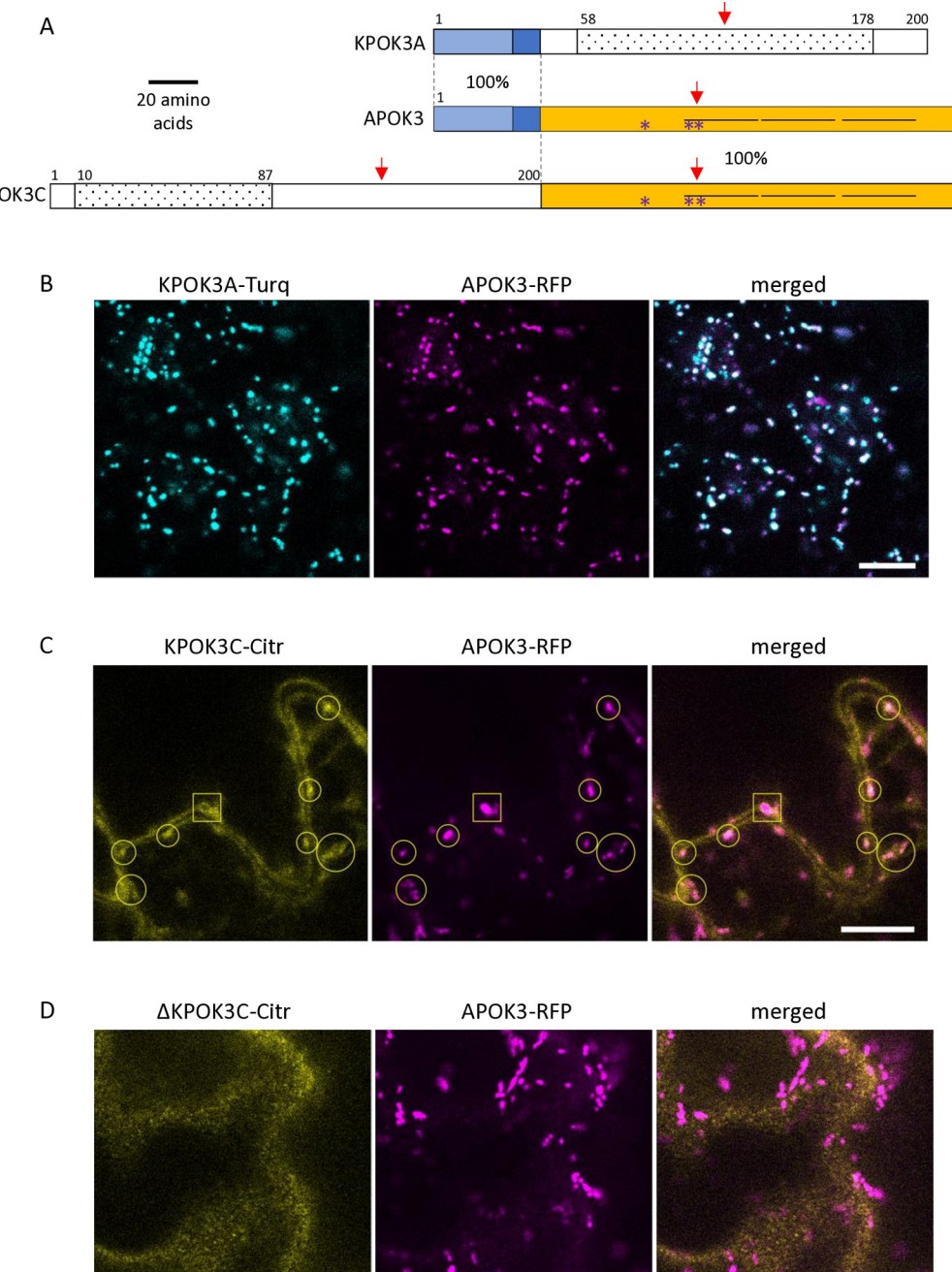

**Fig 4. Structure and cellular localization of the killer proteins. A**. Schematic representation of the two Mr-0 killer proteins compared to APOK3. The homologous regions between the proteins are represented by identical colored boxes with the percentages of identity indicated between proteins within the dotted lines. APOK3 represents the two identical copies of the antidote present in Mr-0. Light blue: 32-amino acid mitochondria-targeting peptide. White boxes represent protein regions with no sequence homology between them. The three HEAT motifs present in APOK3 and KPOK3C are represented by horizontal lines. Dotted boxes indicate predicted NYN domains. Red arrows show the positions of the CRISPR-Cas9 induced mutations. Purple stars indicate the positions of the residues specific for the antidote form of APOK3. **B.** Mitochondrial colocalization of KPOK3A fused to mTurquoise2 (KPOK3A-Turq, cyan) with APOK3 fused to RFP (APOK3-RFP, magenta). **C.** Localization of KPOK3C fused to citrine2 (KPOK3C-Citr, yellow). The square frames an example of a mitochondria surrounded by KPOK3C-Citr, while open circles frame examples of apparently colocalized KPOK3C-Citr and APOK3-RFP. **D.** Cytosolic localization of KPOK3C deprived of its N-terminal part, fused to citrine2 (ΔKPOK3C-Citr, yellow): the citrine2 fluorescence is not associated to RFP-marked mitochondria. Fluorescence was assessed in leaf epidermal cells. Brightness and contrast have been adjusted for clarity. Scale bars: 10 μm.

with the coding sequence of the Citrine2 fluorescent protein. We never observed the fluorescence of the KPOK3C-citrine fusion in the nucleus. Instead, it was observed partly in the cytosol, but also colocalized with APOK3 in the mitochondria (circles in Fig 4C). In some cases, the KPOK3C signal was seen surrounding the APOK3 signal (square in Fig 4C), suggesting that KPOK3C could be located at the mitochondrial periphery. In contrast, when we used a construct where the protein KPOK3C was deleted for its first 116 amino acids, the fluorescent fusion protein was observed only in the cytosol and never associated with mitochondria (Fig 4D), revealing a role of the N-terminal part in mediating KPOK3C localization.

## The killer activity is pollen specific

In the course of the CRISPR-Cas9 mutagenesis of *KPOK3C* with the guide-RNA gRNA_A-POK3-L#2, located in the region identical between *KPOK3C* and *APOK3* (Fig 2B), we obtained in the genotype ShaL3$^M$ some KO mutants in *APOK3-1* and/or *APOK3-2*, the two copies of the antidote gene present in Mr-0, which were not mutated in *KPOK3C*. We used these mutants to validate the functionality of each Mr-0 antidote copy, and to observe the effect of a killer allele devoid of antidote on the plant phenotype. We crossed each of the simple mutants *apok3-1* or *apok3-2* with the killer accession Ct-1 and observed no bias in the F2 progenies (Fig 5B and S4 Table), indicating both Mr-0 *APOK3-1* and *APOK3-2* are able to produce the antidote and each is sufficient to protect pollen grains against the killer. The double mutant homozygous *apok3-1 apok3-2* had no alive pollen (Fig 5A) and was sterile, indicating that the inactivation of both genes turned the Mr-0 allele into a 'suicidal' allele, with a killer activity and no antidote. The killer activity was confirmed by the sterile phenotype of the hybrid between this mutant and Sha (Fig 5A). The loss of the antidote function was confirmed in the

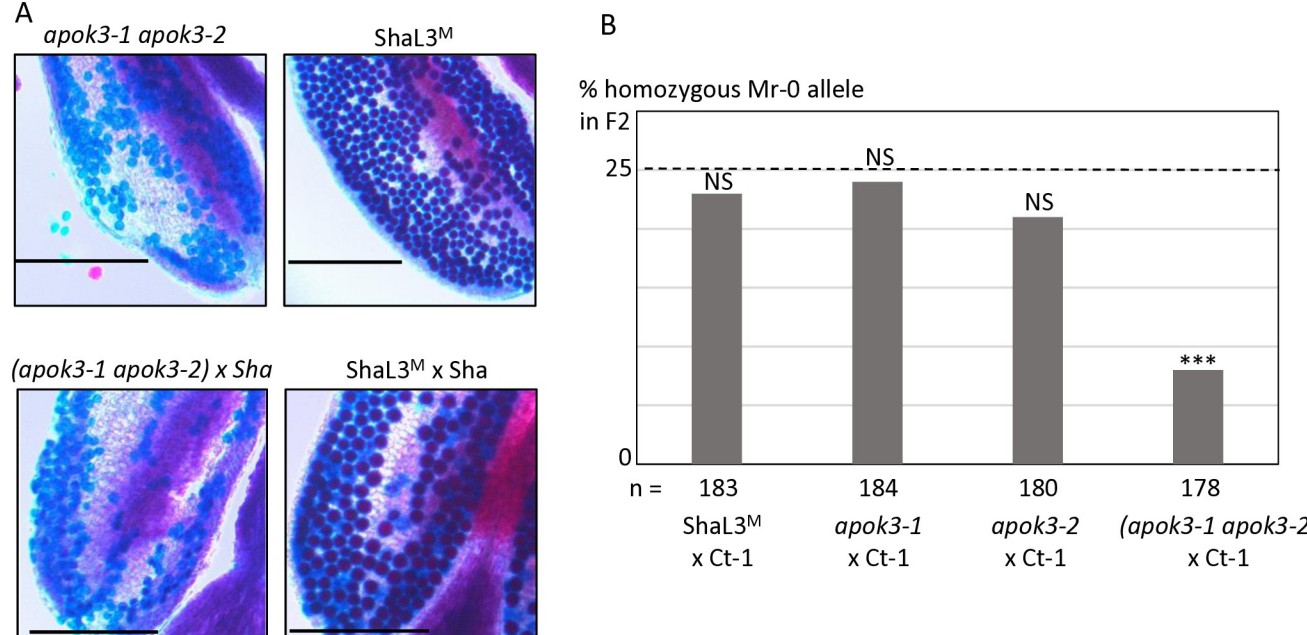

**Fig 5. KO mutations in both antidote genes turns the Mr-0 allele into a suicidal allele. A.** Pollen viability of the double mutant *apok3-1 apok3-2* compared to the wild-type (ShaL3$^M$), and of the hybrid between this mutant and Sha compared to the wild-type hybrid (ShaL3$^M$ x Sha). Scale bars: 200μM. **B.** Percentages of plants homozygous for the Mr-0 allele at the PK3 locus in selfed progenies of hybrids between Ct-1 and plants with the Mr-0 allele, either wild-type or with one or both *apok3* KO mutations. The percentage expected in the absence of segregation distortion (25%) is indicated by a dotted line. n: number of F2 plants genotyped for each cross. *** p < 0.001; NS, not significant. Complete data are shown in S4 Table.

F2 progeny of the cross between the double mutant and the killer accession Ct-1: a bias was observed at the PK3 locus against the Mr-0 double mutant allele (Fig 5B). Remarkably, the double mutant homozygous *apok3-1 apok3-2* presented a normal vegetative phenotype, and was female fertile (as indicated by the production of hybrid seeds when pollinated with Ct-1 or Sha). This indicates that the killer effect of PK3 is specific to pollen.

### An amino acid crucial in APOK3 appears essential in KPOK3C

During the CRISPR-Cas9 mutagenesis with gRNA_APOK3-L#2, we also obtained a plant with a homozygous 3-bp (GAC) deletion 855 bp downstream of the ATG starting position in *KPOK3C* (*kpok3c#3*, Fig 2B). This in-frame deletion led to the amino acid change R262S and the loss of R263, at the beginning of the first HEAT motif of the protein. Because the guide-RNA also targeted the two antidote genes of Mr-0, this mutant was also heterozygous for a KO mutation in *APOK3-1* and homozygous for a KO mutation in *APOK3-2*. Therefore, this mutant possessed only one functional copy of the antidote. It presented no phenotypical alteration compared to the wild-type and was fertile. We used this genotype to test if the mutation in *KPOK3C* affected its killer function by analyzing its selfed progeny. Among 24 plants of the progeny, no bias against the *apok3-1* allele was observed: we obtained 5 plants homozygous for the *APOK3-1* allele, 7 plants homozygous for the *apok3-1* allele, and 12 plants heterozygous. Moreover, all 24 plants were fertile, including the plants homozygous *apok3-1*, *i.e.* totally devoid of antidote. Therefore, pollen grains without functional antidote were not killed: this mutation in *KPOK3C* suppressed the killer activity. Interestingly, the missing amino acid R263 corresponds in APOK3 to R105, the third of the three amino acids specific for the functional antidote form [50] (Fig 4A). This indicates that the same amino acid is essential in both the antidote and killer proteins.

### Behavior of an intermediate form of the antidote

We have previously identified in *APOK3* three SNPs (A/T, A/T and C/T) causing the amino acid changes S85C, D101V and R105C that differentiate a functional (SDR) from a non-functional (CVC) form of the protein [50]. The resistant form of the gene was thus referred to as 'AAC' while the sensitive one was referred to as 'TTT'. Another *APOK3* form, 'TAC' (CDR), was absent from the accessions tested for the antidote function, but was found to be rather common amongst worldwide accessions when the 1001Genomes data were explored [50]. Interestingly, this form possesses a combination of the nucleotides specific to sensitive (TTT) and resistant (AAC) forms. To determine its efficiency as an antidote, 14 accessions with this form of APOK3 were tested in crosses with Mr-0, six belonging to the worldwide collection and eight originating from local French populations (see below). The segregations in the corresponding F2s showed different behaviors among these accessions: while a majority were resistant, a slight bias was detected in the remaining F2s, albeit less pronounced than in the Mr-0 x Sha cross (Fig 6 and S5 Table). We concluded that the TAC form of *APOK3* has an antidote capacity that is however dependent on the genetic backgrounds. Because this TAC form can be poorly efficient, likely because of the S85C substitution in the protein, it was named "weak resistant".

### PK3 diversity in worldwide natural accessions

In addition to Mr-0, we had previously tested the behavior of 18 natural accessions regarding their killer activity, by crossing them to Sha and genotyping the hybrid progenies at the PK3 locus [50]. Here we added 18 new crosses (S6 Table). In total, 13 accessions were found to be killer, and 23 non-killer. We then checked the presence of the three genes of the pollen killer

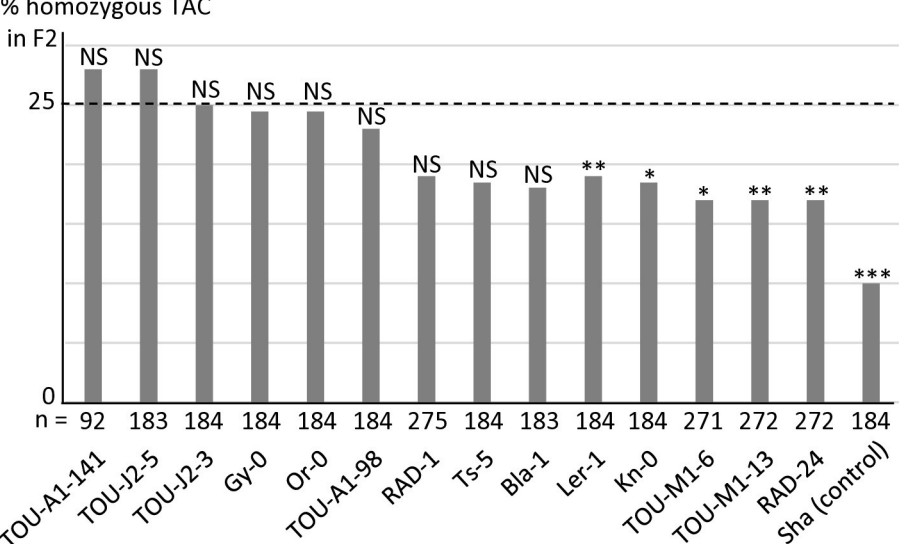

**Fig 6. Behavior of accessions with the TAC form of the antidote in crosses with Mr-0.** Percentages of plants homozygous for the allele with the TAC form of *APOK3* in selfed progenies of hybrids with Mr-0. The percentage expected in the absence of segregation distortion (25%) is indicated by a dotted line. Sha is given as a sensitive control (data from S13 Table). n: number of F2 plants genotyped for each cross. *** $p < 0.001$; ** $p< 0.01$; * $p < 0.05$; NS, not significant. Complete data are shown in S5 Table.

and sequenced them in all these accessions. We defined different forms of the three genes. *KPOK3A* could be absent or truncated, or exist in three main forms, the Sha and Mr-0 forms described above and the Col-0 form, which has the 11 bp located 60 bp upstream of the ATG as Sha but lacks the insertion of 1308 bp in the intron present in Sha (Fig 2A). *KPOK3C* can be present, as in Mr-0, or absent, as in Sha or Col-0. *APOK3* can be absent or take one of the forms AAC, TAC, or TTT, respectively resistant, weak resistant, or sensitive. All the 13 accessions with a killer behavior had the same haplotype as Mr-0 –both a Mr-like *KPOK3A* and a *KPOK3C* gene, with the AAC form of the antidote (Table 3)–, which we thus defined as the killer haplotype, although this haplotype was also observed in 9 accessions which did not kill Sha alleles.

We then sought to obtain a wider view of the PK3 variation within the species. For that purpose, resequencing data from Arabidopsis 1001 Genomes (https://1001genomes.org/) could be used for *APOK3* [50], but they were found to be very incomplete for *KPOK3A*, and do not exist for *KPOK3C* because the gene is absent from the Col-0 reference sequence. We therefore exploited recently published *de novo* genomic sequences [54,55], and investigated 659 accessions of the Versailles Arabidopsis Stock Center (VASC) worldwide collection (https://publiclines.versailles.inrae.fr/catalogue/accession), making a total of 728 accessions (some accessions were present in more than one study) (S7 Table). The VASC collection was analyzed for the presence and the form of each of the three PK3 genes. The *KPOK3A* form (Sha-like, Mr-like, or Col-like) was identified by PCR based on the two characteristic indels that distinguish them. The presence of *KPOK3C* was determined by specific PCR. The *APOK3* form (AAC, TAC, or TTT) was determined by sequencing. The frequencies of the different forms of the genes in the whole set of 728 accessions are shown in Fig 7A. *KPOK3A* was present in 95% of the accessions, and its predominant form was Sha-like (68% of accessions), with the Mr-like and Col-like forms accounting for 14% and 13% respectively. *KPOK3C* was present in only 11% of the accessions. When looking at the four diversity groups identified by Simon *et al.*

**Table 3. PK3 haplotypes of 36 accessions and their behavior in cross with Sha.**

| Accession | *KPOK3A* | *KPOK3C* | *APOK3* | % Sha [a] | Behavior |
|---|---|---|---|---|---|
| Etna-2 | Mr-like | + | AAC | 2 *** | killer [b] |
| Jea | Mr-like | + | AAC | 3 *** | killer [b] |
| Lom-1 | Mr-like | + | AAC | 3 *** | killer |
| Toufl-1 | Mr-like | + | AAC | 3 *** | killer |
| Eden-2 | Mr-like | + | AAC | 4 *** | killer |
| Toum-3 | Mr-like | + | AAC | 5 *** | killer |
| Can-0 | Mr-like | + | AAC | 6 *** | killer |
| Cant-1 | Mr-like | + | AAC | 8 *** | killer [b] |
| Shigu-2 | Mr-like | + | AAC | 9 *** | killer [b] |
| Mr-0 | Mr-like | + | AAC | 10 *** | killer [b] |
| Ct-1 | Mr-like | + | AAC | 12 *** | killer [b] |
| TOU-A1-111 | Mr-like | + | AAC | 15 *** | killer |
| TOU-A1-74 | Mr-like | + | AAC | 17 ** | killer |
| N7 | Mr-like | + | AAC | 18 NS | non-killer |
| Esp2-1 | Mr-like | + | AAC | 21 NS | non-killer |
| Hel-1 | Mr-like | + | AAC | 23 NS | non-killer |
| N2 | Mr-like | + | AAC | 25 NS | non-killer |
| Etn-0 | Mr-like | + | AAC | 28 NS | non-killer [b] |
| Had-6b | Mr-like | + [c] | AAC | 21 NS | non-killer |
| Kas-1 | Mr-like | + [c] | AAC | 29 NS | non-killer [b] |
| Lov-5 | Mr-like | - | AAC | 23 NS | non-killer [b] |
| Bur-0 | Mr-like | - | AAC | 28 NS | non-killer [b] |
| Ru3,1–27 | Sha-like | + | AAC | 22 NS | non-killer |
| Uk-2 | Sha-like | + | AAC | 23 NS | non-killer |
| Are-10 | - | - | TTT | 20 NS | non-killer [b] |
| Cvi-0 | - | - | TTT | 27 NS | non-killer [b] |
| Ita-0 | Col-like | - | - | 26 NS | non-killer [b] |
| Lou-9 | Col-like | - | AAC | 28 NS | non-killer |
| Col-0 | Col-like | - | AAC | 30 NS | non-killer [b] |
| Sorbo | Sha-like | - | TTT | 21 NS | non-killer [b] |
| Kz-9 | Sha-like | - | TTT | 25 NS | non-killer [b] |
| Qar-8a | Sha-like | - | TTT | 27 NS | non-killer [b] |
| TOU-A1-98 | Sha-like | - | TAC | 23 NS | non-killer |
| Oy-0 | Sha-like | - | AAC | 21 NS | non-killer [b] |
| Lou-17 | Sha-like | - | AAC | 26 NS | non-killer |
| Blh-1 | Sha-like | - | AAC | 28 NS | non-killer [b] |
| Tsu-0 | Sha-like | - | AAC | 28 NS | non-killer |

[a] % homozygotes Sha at the PK3 locus in F2 $p$ $\chi^2$ (Sha:Hz:notSha = 1:2:1): *** $p < 0.001$; ** $p < 0.01$; NS, not significant.

[b] from [50]

[c] *KPOK3C* not in *AT3G62610*

[56], these frequencies were roughly similar in clusters 1 (Russia and Central Asia), 3 (Germany and Western Europe), and 4 (France, Spain, Italy). However, in cluster 2 that groups Nordic and Mediterranean accessions, the Mr-like form of *KPOK3A* as well as the presence of *KPOK3C* were increased respectively to 46% and 29% (S7 Table). Globally, the two main forms of *APOK3*, which was present in 92% of the accessions, were the weak resistant form

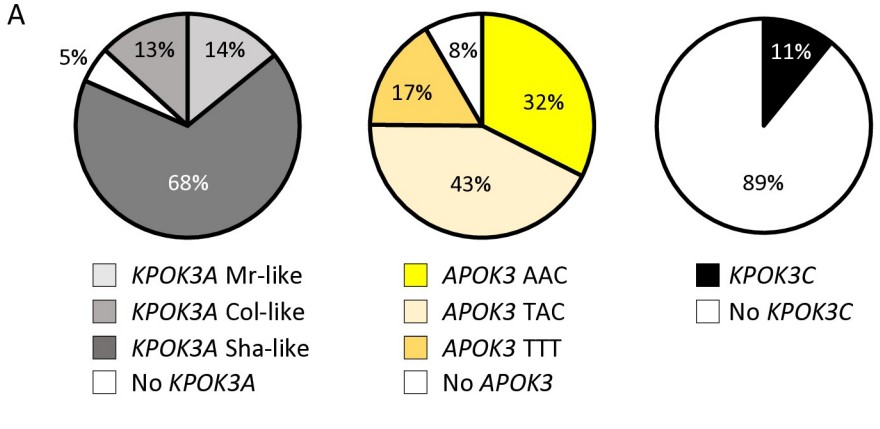

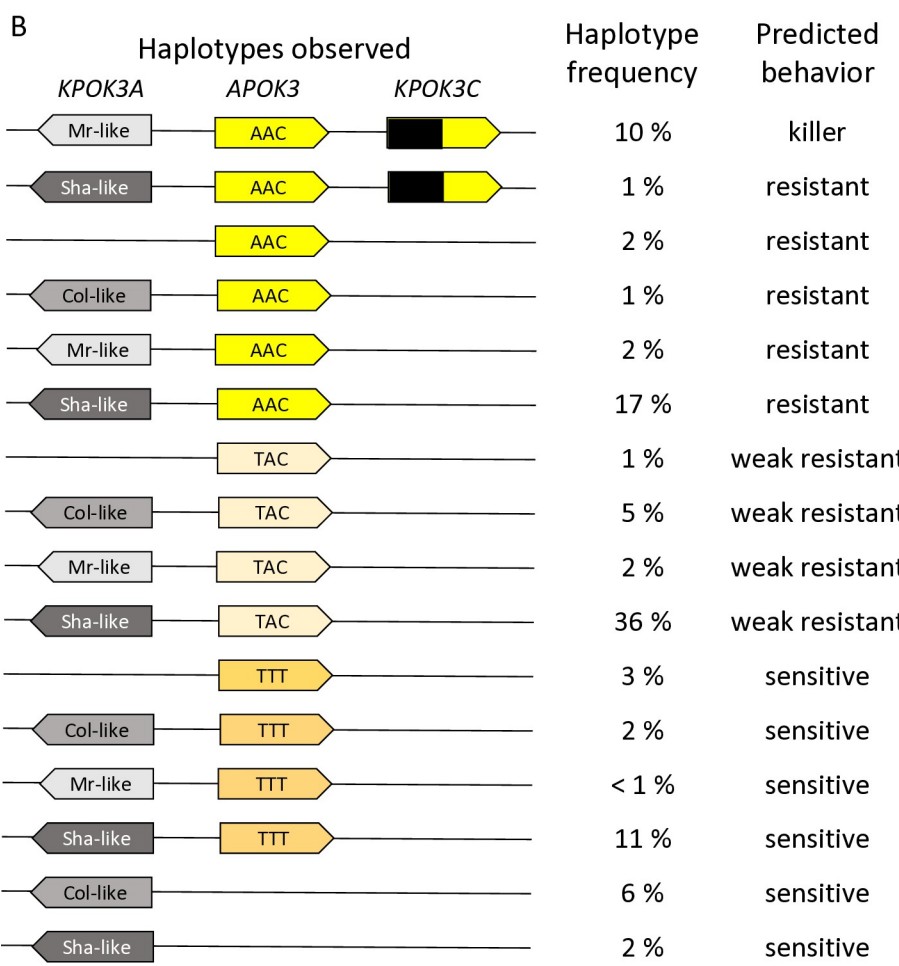

**Fig 7. Global diversity of the PK3 alleles among 728 natural accessions. A.** Distribution of the different forms of the three PK3 genes. **B**. Simplified haplotypes observed, with their frequencies among the 728 accessions and predicted behaviors. Complete data are presented in S7 Table.

TAC (43%) and the resistant form AAC (32%) (Fig 7A). However, the sensitive TTT form was present in 77% of the accessions from Russia and Central Asia (cluster1) (S7 Table).

Analysis of the PK3 locus structure in the 131 newly sequenced accessions [54, 55] revealed new structural variants, confirming its hyper-variable nature [50]. While most of the 113 accessions with non-killer haplotypes had no or only one *APOK3* copy, 17 had several copies (AAC, TAC, or TTT), most often 2 or 3, but up to 10 in the accession IP-Fel-2. In addition, in the accession Rld-2, two identical copies of *KPOK3C* are present in *AT3G62610*. In three accessions (Had-6b, Kas-1 and Ms-0), *KPOK3C* is not inserted in *AT3G62610* but elsewhere in the locus. In total, this was observed in 20 of the 79 accessions that possess a *KPOK3C* gene in the whole collection (S7 Table).

To try to get insights into the molecular evolution of the PK3 genes, we retrieved the genomic sequences for all *KPOK3A*, *APOK3*, and *KPOK3C* genes from the 131 accessions and aligned them. This analysis showed that *KPOK3A* sequences presented many non-synonymous SNPs in addition to the indels used to differentiate Sha-like, Mr-like and Col-like forms (S8 Table). In contrast, *KPOK3C* sequences were very conserved (S9 Table), and, in the region homologous to *APOK3*, the three nucleotides that characterize the functional form of *APOK3* were all of the resistant AAC form. Alignment of *APOK3* sequences showed that, in the 34 accessions with two or three copies of *APOK3*, the paralogs are identical, except for BARC-A-12, BARC-A-17 and Bur-0, where one of the three copies differs from the two others by a unique SNP, without changing the functional form (S10 Table). In IP-Fel-2, four of the 10 *APOK3* copies are identical while the others all differ from each other. Interestingly, three of these copies possess a novel TAT form of the gene, which encodes a new protein type, CDC (S10 Table). Sanger sequencing of the accessions from the VASC collection revealed that some other accessions likely possess several copies of *APOK3* with two different forms, TAC and this novel TAT form, or TAC and TTT (S7 Table). In IP-Mos-9, which has a TTT form of *APOK3*, the presence of an additional SNP next to the first T results in another new form (SVC) of the protein (S10 Table). The phylogenetic tree inferred from the *APOK3* nucleotide sequences (S4 Fig), which could not be rooted with an outgroup sequence because the gene is species-specific, showed that the TAC form is partitioned into two distinct clades, due to a SNP (C/T) located 36 bp before the ATG starting position (S10 Table). All the AAC forms of *APOK3* had a C at this position, while most (23) of the TTT ones had a T, seven a G and two a C. As a consequence, each of the two other forms is closer to one of the two TAC clades.

The phylogenetic tree of *KPOK3A* nucleotide sequences grouped them according to the forms we have defined on the basis of indel polymorphisms, except that the Col-like sequences were split into two clades (Fig 8). If we examine the protein polymorphism data (S8 Table), one of these clades is very close to the Mr-like form while the other shares more variant residues with the Sha-like form. In addition, the African accession Tanz-1 displays all the amino acid changes characteristic of the Sha-like form. This explains its presence at the basis of the Sha-like clade despite being typed as a Col-like sequence based on the indels (Fig 8). Altogether, these results suggested that the Sha-like and Mr-like forms derived independently from two types of the Col-like form of *KPOK3A*, which seems to be the ancestral form.

In the whole set of 728 accessions, when considering the different forms of the three PK3 genes together, we observed 16 different haplotypes out of the 32 theoretically expected, with contrasting frequencies (Fig 7B). Indeed, when *KPOK3C* was present, *APOK3* was always of the resistant form AAC, and 94% of these accessions had the Mr-like form of *KPOK3A*. This haplotype, the only one capable of providing killer activity, concerned 10% of all the accessions studied, and it was mainly present in cluster 2 (Nordic and Mediterranean accessions), where it accounts for 30% of the accessions. The three most represented haplotypes had a Sha-like *KPOK3A*, no *KPOK3C*, and a form of *APOK3* either TAC, AAC, or TTT (36%, 17%, and 11% of the accessions, respectively), while 10 haplotypes were represented in less than 5% of the accessions. The behavior of non-killer haplotypes can be predicted after their *APOK3* form:

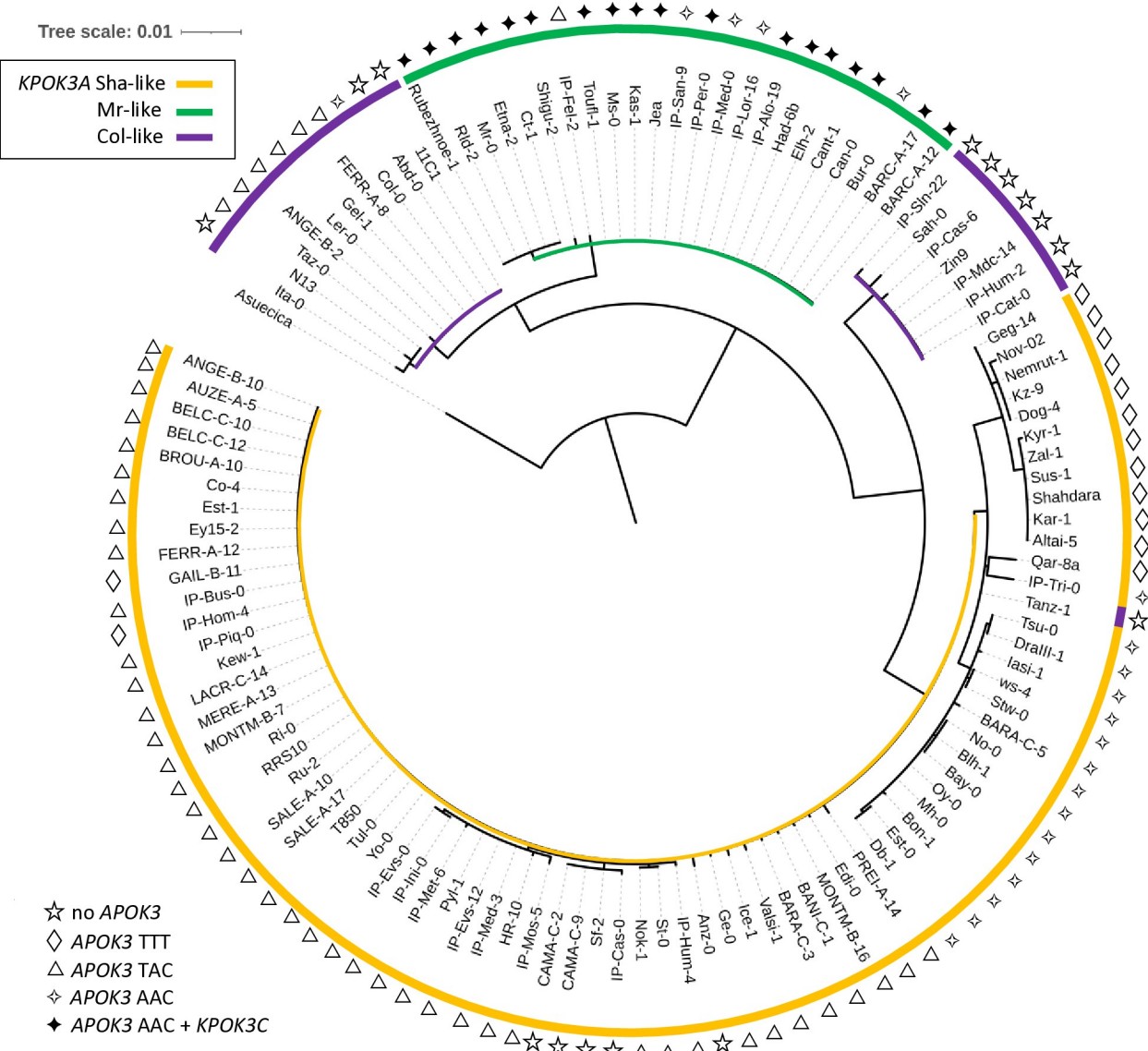

**Fig 8. Phylogenetic tree of *KPOK3A* sequences.** The sequence from *A. suecica* was used as an outgroup. Colors indicate the type of *KPOK3A* as defined by the indels in the promoter and intron. The symbol at each leaf indicates the associated functional type of *APOK3*: sensitive (TTT), weak resistant (TAC), or resistant (AAC); filled symbol indicates the presence of *KPOK3C* in the accession.

those with the AAC form (25%) are expected to be resistant, while those with the TTT form (18%) or without *APOK3* (9%) are expected to be sensitive. Predicting the behavior of accessions with the TAC form (48%) is more problematic, as shown above, yet we classed these haplotypes as weak resistant. The geographic distribution of the predicted behaviors showed that killer and sensitive genotypes are both present in Europe at the regional scale (Fig 9). This suggested that they can coexist in local populations.

## Killer and sensitive haplotypes coexist in local natural populations

We then focused on the French TOU-A population, collected along a 330-meter transect under an electric fence separating two permanent meadows in Burgundy [57]. This local population

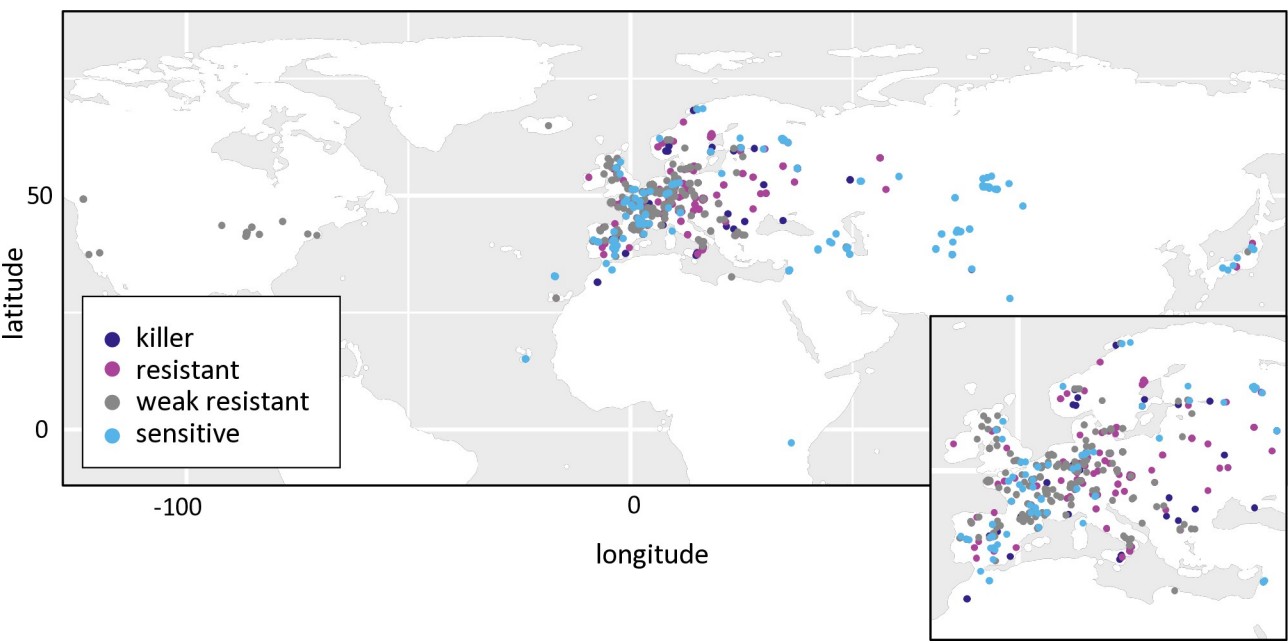

**Fig 9. Geographic distribution of predicted PK3 behaviors in the global collection.** Predicted behaviors of the accessions (S7 Table) were plotted on the world map in R using the ggplot2 package and its world shape file as the basemap. Each dot represents one accession, with predicted behaviors indicated by the dot color. Europe is enlarged in the inset for better resolution.

has been demonstrated to by highly polymorphic at both the genomic and phenotypic levels [57]. We investigated 301 TOU-A accessions for their *KPOK3A* and *APOK3* forms and for the presence of *KPOK3C*, and found that killer and sensitive haplotypes were present in the population (Fig 10 and S11 Table). The killer haplotype was much more represented than in the worldwide collection: 155 of 301 accessions had both a Mr-like form of *KPOK3A* and a gene *KPOK3C*, plus an AAC form of *APOK3*. Among the 146 accessions with non-killer haplotypes, 107 had a TAC form of *APOK3*, and the 39 remaining accessions had sensitive haplotypes, 12 accessions with a TTT form of *APOK3* and 27 accessions with no *APOK3* (S11 Table).

We then searched whether killer and sensitive haplotypes also coexisted in 11 other local populations from Burgundy [48] (Fig 10). Ten to 20 accessions collected randomly from each stand were investigated first for the presence of *KPOK3C*. This gene was found in only 3 of these populations, the most geographically close to the TOU-A population (Fig 10 and S12 Table). In these three populations, we then determined the allelic forms of *KPOK3A* and *APOK3*. In the RAD and TOU-M1 populations, non-killer haplotypes, either with a sensitive or a TAC form of the antidote, were also present, whereas no sensitive haplotype was found in TOU-J2 (Fig 10 and S12 Table).

We crossed plants with killer haplotypes from the TOU-A, TOU-M1, TOU-J2, and RAD populations with Sha in both ways and tested the allelic segregation at the PK3 locus in the hybrid progenies. A bias against the Sha allele was found in every progeny, although the strength of the bias depended on the accession tested (Fig 11A and S13 Table). The bias was stronger when Sha was the female parent of the crosses, as previously reported for the hybrids between Sha and Mr-0 [50]. Similarly, representative plants of each population with sensitive haplotypes (a TTT form of *APOK3* or no *APOK3*) were crossed with Mr-0. As expected, genotyping the progenies of the hybrids at the PK3 locus showed that all these plants had a sensitive behavior (Fig 11B and S14 Table). Killer alleles present in the Burgundy populations were thus active against Sha, and sensitive alleles were killed by Mr-0.

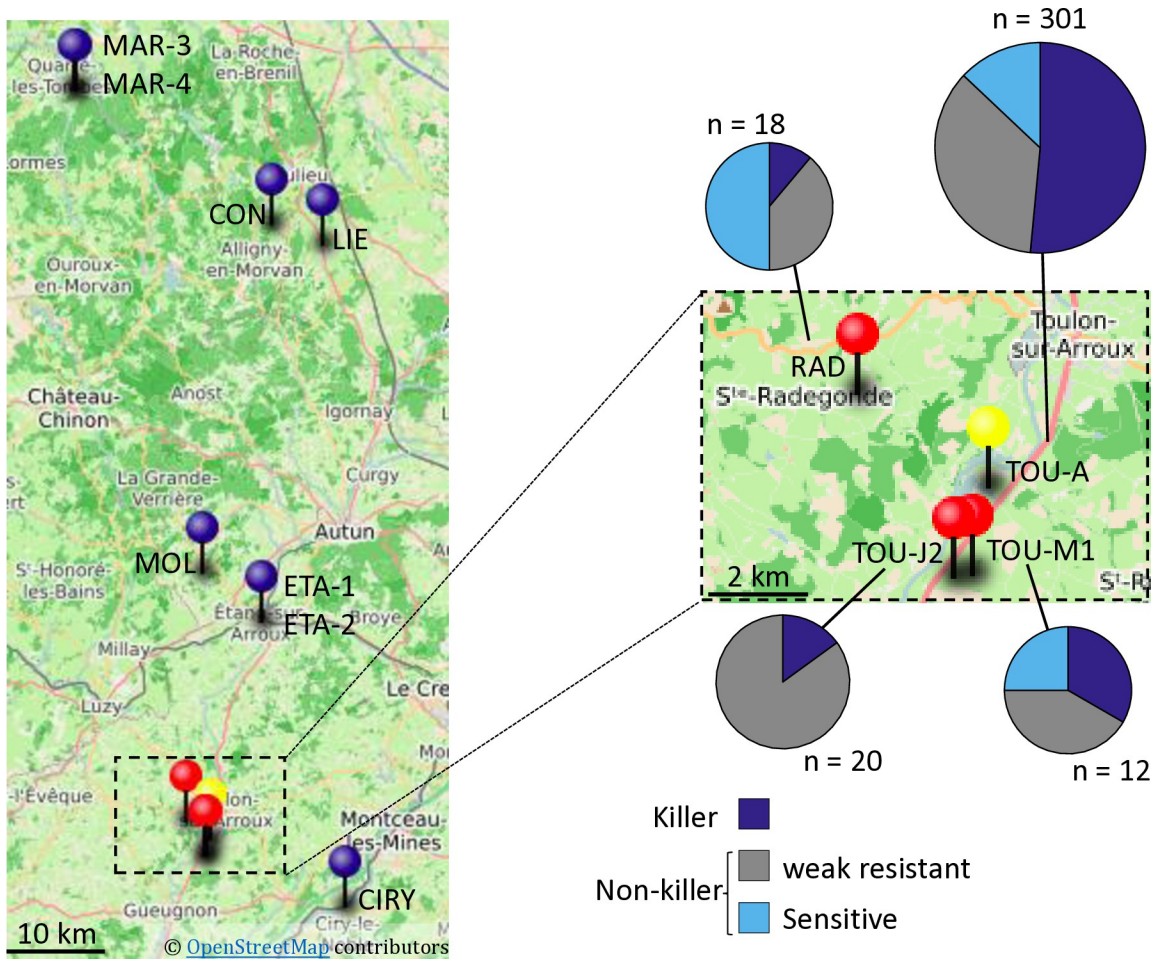

**Fig 10. PK3 in local Burgundy populations. A.** Geographical location of the populations studied. Blue pins: populations without killer haplotypes; red and yellow (TOU-A) pins: populations with both killer and non-killer haplotypes. The map was created using uMap (https://umap.openstreetmap.fr/en/about/); GPS data are from Brachi *et al.* [48]. A close up on the populations where killer haplotypes were found is shown on the right, with pie charts representations of the distribution of PK3 haplotypes within the populations (Complete data in S11 and S12 Tables).

We then analyzed several crosses between killer (with respect to Sha) and sensitive (with respect to Mr-0) accessions from the Burgundy populations. The strength of the bias differed greatly according to the cross (Fig 11C and S15 Table). The observed segregations resulted from both the strength of the killer parent and the sensitivity level of the sensitive parent. Some crosses between a killer and a sensitive accession showed no significant segregation distortion in their progenies, in particular hybrids between TOU-A parents, between TOU-J and TOU-A parents, or between TOU-J and TOU-M parents. Indeed, the accessions TOU-A1-111 and TOU-J2-10 were weak killers, able to kill only Sha and RAD-8 sensitive alleles. In contrast, the killer effect of RAD-2 was very strong. It is noticeable that the RAD population, which contains this strong killer accession, also contains a very sensitive one (RAD-8). Therefore, the pollen killer could be active in this population, and cause the spread of the killer haplotype.

## Discussion

In this study, we first complete the genetic characterization of the pollen killer PK3 by identifying the two killer genes, *AT3G62460*, named *KPOK3A*, and a novel gene present only in certain

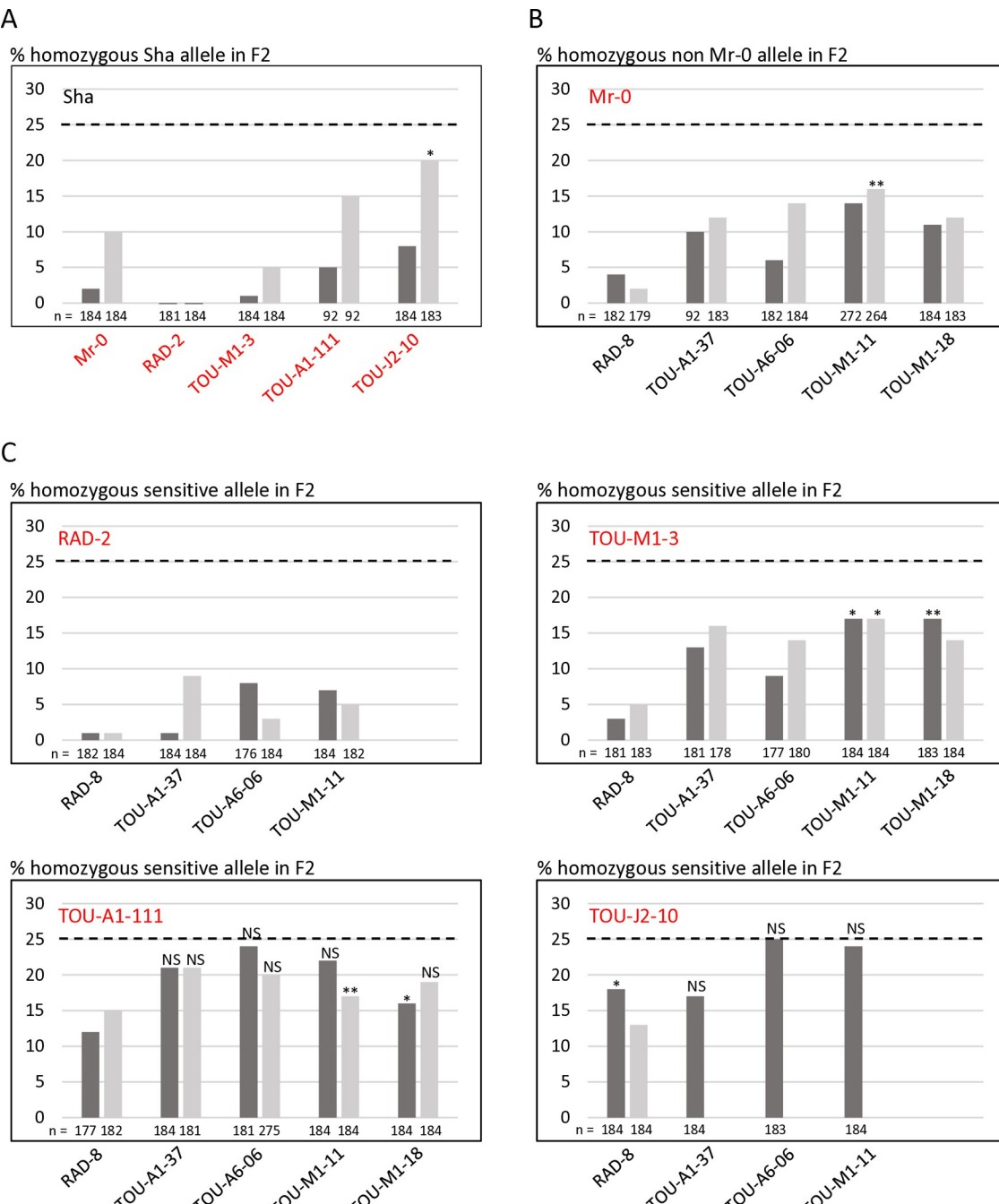

**Fig 11. PK3 behavior in Burgundy accessions. A**. Validation of the behavior of killer alleles from Burgundy accessions in crosses with Sha. **B.** Validation of the behavior of sensitive alleles from Burgundy accessions in crosses between accessions of local populations carrying sensitive and killer alleles. **C.** PK3 effect in crosses between accessions of local populations carrying sensitive and killer alleles. Each panel shows the percentages of plants homozygous for the sensitive allele in the progenies of reciprocal crosses between accessions carrying sensitive and killer alleles (dark grey, the sensitive accession is the female parent; light grey, the sensitive is the male parent). The percentage expected in the absence of segregation distortion (25%) is indicated by a dotted line on each panel. The names of the accessions with killer and sensitive alleles are written in red and in black, respectively. n: number of F2 plants genotyped for each cross. All segregations significantly differ from the expected ones with a *pvalue* < 0.001, except those marked with * (*p* < 0.05), ** (*p* < 0.01), or NS (not significant). Complete data are shown in S13–S15 Tables.

accessions, *KPOK3C*. These genes are located respectively in the PK3A and PK3C intervals, flanking the antidote gene *APOK3* previously identified [50], and they are both required for the killer function, which complicates the deciphering of the pollen killer mechanism. To our knowledge, only two gamete killers with more than one killer gene have been reported in plants so far, both in rice crosses. In the S5 female distorter, one of the killer genes encodes an aspartic protease and the other a protein with no homology with any known protein [23]. In the S1 male and female gamete killer, two killer genes likely evolved in ancient wild rice species, one encoding a small peptidase with similar features to the protector - which is also required for the killer function - and the other encoding an uncharacterized protein with no putative domain identified [25]. In none of these two gamete killers, as in our case, has the molecular mechanism leading to gamete abortion been elucidated to date. The genes involved in eukaryotic poison-antidote systems are generally species-specific novel genes [34]. So, although many such systems have been identified, their molecular mechanisms remain largely unknown.

As for some other pollen killers [24,26,30], the killer effect of PK3 was limited to pollen, since the double mutant *apok3-1 apok3-2* of Mr-0, devoid of antidote, was indistinguishable from the wild-type, except it was male sterile (Fig 5A). The PK3 behaves as a poison-antidote system [50], which implies that the killer genes are expressed at the sporophytic stage, at least in meiocytes, so that all pollen grains receive the killer proteins. For instance, in *Schizosaccharomyces pombe* Wtf spore killer, the poison is specifically produced in meiocytes [58]. We detected *KPOK3A* and *KPOK3C* expression in sporophytic tissues (Figs 3B, 3C and S3), and it is conceivable that the production of killer proteins is enhanced in meiocytes, as they are expected to remain in pollen grains after meiosis. In such case, the weak expression of *KPOK3C* in buds and leaves (Figs 3C and S3B) could explain the pollen specificity of the killer: it is possible that expression in other tissues would be too low to induce a toxic effect. This could also explain why the *KPOK3C* genomic sequence containing 650 bp of promoter sequence, although expressed at a high level in buds (Fig 3G), was unable to complement the mutant: the promoter sequence used would lack an element required for sufficient *KPOK3C* expression in meiocytes. The fact that the pRPS5A promoter, expressed in meiocytes, is also unable to complement the mutant is in favor of the need for a precise expression pattern of *KPOK3C*. Another possible, non-exclusive explanation for the pollen-limited phenotype of the double mutant *apok3-1 apok3-2* of Mr-0 would be that the toxicity is specific for pollen grains. It is the case in the recently reported *Teosinte Pollen Drive* of maize, where *Tpd1* encodes a long non-coding RNA producing si-RNAs that specifically target a gene encoding a lipase required for pollen viability [30]. Pollen specificity is also observed in cytoplasmic male sterilities (CMS), where it can be ensured either by the pollen- or tapetum-specific production of mitochondrial sterility factors or by the triggering of a pollen-specific cell death mechanism [59–61]. Interestingly, direct interaction with OsCOX11, a subunit of the mitochondrial CYTOCHROME OXIDASE, was reported to trigger pollen death in both the rice Wild Abortive CMS [62] and the *RSH12* pollen killer, acting in japonica/indica rice hybrids [27]. Neutralization of the PK3 killer activity likely occurs in mitochondria, where APOK3 and KPOK3A co-localize (Fig 4B) due to their identical mitochondrial targeting sequences. The preferential association of KPOK3C with mitochondria (Fig 4C) supports that mitochondria are also the site of the killer activity. Intriguingly, KPOK3C was sometimes observed at the mitochondrial periphery, which is reminiscent of cytosolic glycolytic enzymes that are associated with mitochondria in response to respiratory demand [63]. A more precise investigation of its association with mitochondria (outside/inside, direct/indirect), and the involvement of its first 116 residues in this association (Fig 4D), will help to better understand the role of KPOK3C. Notably, it would be interesting to investigate whether the association of KPOK3C with

mitochondria is linked to that of glycolytic enzymes, or whether it impacts the response to respiratory demand. The implication of mitochondria in the PK3 functioning is also supported by the fact that the cytoplasmic background of hybrids may affect the strength of the bias, as previously observed in reciprocal crosses between Sha and Mr-0 [50], and here in reciprocal hybrids with several Burgundy accessions (Fig 11). Mitochondria involvement has been reported in other gametes killers. For instance, in the rice *qHMS7* pollen killer locus, the anti-dote is localized in mitochondria [24], and, in the Wtf spore killer of fission yeast, the killer activity is affected by genes involved in mitochondrial functioning [64].

Remarkably, although their primary structures are not homologous, the two killer proteins KPOK3A and KPOK3C both contain an NYN domain of the LabA-like PIN domain of lim-kain-b1 type. Another type of PIN-like domain, PIN.10, has been described as a feature of bacte-rial toxin-antitoxin systems [65]. Considering that PIN-like domains constitute a superfamily of nucleases, it is relevant to note that nuclease activity was shown to be responsible for the cell toxicity of the *Podospora anserina* spore killer SPOK1 [66]. However, the NYN.1 cluster, to which belong the NYN domains of KPOK3A and KPOK3C, has not been assigned any putative function yet [65]. So, the production and functional analysis of mutants driven by the NYN domain consensus structure in both killer proteins would provide insights into their mode of action.

In several fungal spore killers, the toxin and antidote present sequence homologies because they are produced by the same gene or very close paralogues [58,67–72]. Both PK3 killer pro-teins share a part of their sequence with the antidote (Fig 4A). Besides the sequence identity between APOK3 and KPOK3A over their first 44 amino-acids that include the mitochondria-targeting peptide, APOK3 and KPOK3C share their 179 C-terminal residues. This encom-passes almost the entire mature antidote protein, including three HEAT-repeats and the three residues that distinguish functional (SDR) and non-functional (CVC) forms of the antidote (Fig 4A). HEAT repeat domains are predicted to mediate protein-protein interactions [73] and their role in possible interactions between killer and antidote proteins or with other pro-teins, notably mitochondrial proteins, needs to be investigated. Our results on accessions with the CDR form of APOK3 (encoded by the TAC form of the gene) indicate that the S85C change influences the antidote efficiency (Fig 6). This is coherent with the fact that antidotes of killer alleles are always of the AAC form, which is the most efficient antidote form observed. Testing the antidote activity of the new protein forms of APOK3 uncovered in this study, CDC and SVC, will provide further insight into the functional roles of these residues. Remarkably, all the *KPOK3C* genes sequenced encode an SDR form of the killer protein. In addition, we have shown that the R residue in the SDR signature of APOK3, at the beginning of the first HEAT repeat, is also important for KPOK3C killer activity. Moreover, among the 18 killer alleles we analyzed, the shared regions of APOK3 and KPOK3C were identical in 14 accessions and differed by a single amino acid in the other four (S9 and S10 Tables). The homology between KPOK3C and APOK3 strikingly reminds of the Wtf spore killer. The two Wtf pro-teins, poison and antidote, are encoded by a single gene through alternative transcription starts, ensuring the identity of the shared region, with the antidote protein having 45 addi-tional N-terminal residues compared to the poison [58]. All the known active Wtf antidotes are highly similar to the poison they neutralize, and mutations that disrupt the similarity can eliminate the ability of the antidote to neutralize the poison through aggregation and cellular relocation [32,64]. To what extent such a homology between APOK3 and KPOK3C is required for the antidote activity in the case of PK3 has to be investigated. Our results provide a basis for exploring the biochemical nature of the interactions between the different components of PK3 and how they differ between alleles.

Our results made it possible to predict the behavior of different PK3 haplotypes. We defined the killer haplotype as possessing a gene *KPOK3C*, an Mr-like form of *KPOK3A*, and the resistant AAC form of *APOK3*, because all 13 killer accessions had these alleles. However, some accessions with this haplotype did not show a killer behavior (Table 3). This could be linked to the location of *KPOK3C* within the locus, as observed for Had-6b and Kas-1 where it is located in a TE-rich region, possibly preventing its correct expression. Otherwise, unlinked modifier loci that modulate the strength of the bias could be present in the genomes of the accessions with unexpected behavior. Such modifiers have been reported in several different distorters [19,31,74,75]. Also, we cannot exclude that an additional gene in the PK3C interval remains uncovered and could be lacking in these accessions.

From the genetic fine mapping of the locus, we know that the *KPOK3A* from Sha has no killer activity [50]. In addition, accessions that possessed a Sha-like *KPOK3A* associated with a gene *KPOK3C* were non-killers (Table 3). Therefore, we assume that Sha-like forms of *KPOK3A* have no killer activity. Yet, this remains to be established for the Col-like forms of *KPOK3A*, particularly those with sequences closer to the Mr-like form (S8 Table). However, *KPOK3A* expression is very weak in both Sha and Col compared to Mr-like forms (Figs 3B and S3A), and we hypothesize that this difference is due to the 11 bp indel located 60 bp upstream of the ATG, just next to a putative TATA box (Fig 3D). This polymorphism would thus contribute to functional differences between Mr-like and the two other forms of *KPOK3A*.

The availability of a large, worldwide collection of natural accessions allowed us to get a picture of the distribution of PK3 haplotypes within the species (Figs 7 and 9). *KPOK3C* and *KPOK3A* were typed according to the structural variations that were associated with the PK3 behavior, and *APOK3* according to the three SNPs that impact the antidote activity. Taking the three genes together, we only observed 16 haplotypes out of the 32 theoretically possible combinations. This can reflect two non-exclusive effects. First, the killer alleles must have a functional antidote, and, unsurprisingly, none of the predicted 'suicidal' haplotypes was observed. Second, genetic linkage strongly limits recombination between the PK3 genes, particularly between alleles differing by structural variations. Indeed, the PK3 locus, where genome collinearity between accessions is greatly affected [50], can be seen as a hot spot of rearrangements. Such loci contain variable copy numbers of genes, show reduced meiotic recombination in hybrids, and likely undergo particular evolutionary dynamics [76]. In this study, we further extend observations of the remarkable structural diversity of the locus, notably by uncovering, thanks to *de novo* sequencing, multiple cases of *APOK3* duplication in nonkillers, including the accumulation of 10 copies in IP-Fel-2, and one case of *KPOK3C* duplication in a killer (Rld-2) (S7 Table). Because duplicated copies of *APOK3* almost always have the same sequence, even more cases of multiple *APOK3* copies certainly remained undetected by Sanger sequencing of the VASC collection. Limited recombination due to structural variation could at least partly explain that ten out of the 16 observed haplotypes were rare (less than 5% of the panel, Fig 7B).

Our findings show that the pollen killer PK3 has emerged within *A. thaliana*. *KPOK3C* and *APOK3* could not be found in any related species. *KPOK3A* orthologs are present in *A. thaliana* closest relatives *A. lyrata*, *A. arenosa* and *A. suecica*. Yet, *KPOK3A* is not essential in *A. thaliana*: 38 out of 728 accessions have no *KPOK3A* or a truncated gene (S7 Table) and, in Mr-0, inactivation of the gene did not affect the phenotype in our standard growing conditions. We propose that the *A. thaliana* specific PK3 genes appeared sequentially. First, *APOK3* would result from the fusion of the 5' part of *KPOK3A* with parts of at least one paralogue of *AT3G43260* [50]. Second, the association of the largest part of *APOK3* with a long 5' extension of still elusive origin would have produced *KPOK3C*. Our phylogenetic analysis of *KPOK3A* sequences on 131 accessions suggest that the Col-like form of *KPOK3A* is ancestral (Fig 8).

Furthermore, it is likely that the derived forms—Sha-like and Mr-like—evolved independently from already divergent types of this ancestral form. Indeed, although *KPOK3A* is the most variable in sequence among the three PK3 genes, very few non-synonymous polymorphisms are shared by the Sha-like and Mr-like forms (S8 Table). It is conceivable that the Sha-like form derived from a *KPOK3A* gene similar to that of Tanz-1, an African relict [55] whose *KPOK3A* form was defined as Col-like on the basis of the two indel polymorphisms but which has all the SNPs of the Sha allele (Fig 8 and S8 Table). One possible scenario is that the killer haplotype resulted from the formation of a *KPOK3C* gene in an accession with an Mr-like *KPOK3A* and an AAC *APOK3*. In any case, the killer activity must have appeared in the presence of an active antidote, obviously because it would not have been maintained in the absence of a functional antidote. This is also indirectly testified by the fact that *APOK3* is the source of a large part of the *KPOK3C* sequence. We conclude that the pollen killer PK3 likely emerged according to the antidote-first model [38]. Still, a puzzling question is raised by the secondary acquisitions of identical polymorphisms in *APOK3* and *KPOK3C* as in Mr-0 and Etna-2 (S9 and S10 Tables).

While these results strongly support that PK3 genes have evolved within the *A. thaliana* species, they did not provide information on a possible activity of the pollen killer at the population scale. In the antidote-first scenario, the distorting activity of the killer haplotype cannot be expressed in populations where the antidote is fixed [38,40]. This may explain that documented gamete killers in plants mainly operate in crosses between species or subspecies, the best examples being the numerous segregation distorters studied in inter-specific and -subspecific rice crosses [26–28,77]. However, this could also be linked to the fact that most of the documented plant gamete killers concern cultivated species, which evolved under artificial selection. Indeed, the female meiotic driver *D* uncovered in crosses between Mimulus species [17,78] was also reported within several populations of *Mimulus guttatus* with different evolutionary histories [79]. A strong cost on pollen fertility was found associated with the female *D* drive locus, and difference in the strength of drive between intra- and inter-specific crosses was proposed to be due to the selection of suppressors in *M. guttatus* [78]. Regarding *A. thaliana* PK3, the geographical haplotype distribution (Fig 9) suggested that the PK3 may be active in natural European populations, provided that killer and sensitive genotypes can mate. Indeed, the uncovering of local Burgundy populations where killer and sensitive alleles coexist suggests that it is active within natural populations. Here we document three populations (TOU-A, TOU-M1 & RAD) with co-occurrence of killer, weak resistant and sensitive haplotypes (Fig 10). Interestingly, these quite closely located populations possess killer and sensitive alleles of different efficiencies when confronted to the Sha and the Mr-0 alleles, respectively (Fig 11A and 11B). Here again, the strength of the pollen killer likely depends on modifiers present elsewhere in the genome. Moreover, these modifiers counteract more efficiently the PK3 effect in crosses between individuals from the same population than in crosses with distant accessions (Sha or Mr-0). This is consistent with a selection of modifiers within populations. These PK3 modifiers could be selected in genomes confronted to killer alleles in response to an associated cost [38]. A recent modeling work based on rice, a mainly inbred species, proposes that the main cost of a pollen killer would be linked to pollen limitation at the flower scale, *i.e.* when the number of viable pollen is lower than the amount needed to fertilize all the ovules [77]. Although we did not observe any limitation of selfing seed set in plants with an active PK3 grown in standard greenhouse conditions, we cannot exclude a cost in nature. Interestingly, while outcrossing is logically required for the spread of a gamete killer, recent modeling works reported that inbreeding favors the spread of killer alleles within and across populations and even subspecies, provided a certain rate of outcrossing [40,77]. The modeling of fungal spore killers also predicted that a high selfing rate is compatible with the spread and fixation of the killer when associated with low fitness cost [80]. Indeed, variable

rates of outcrossing have been reported in *A. thaliana* natural populations [81], and the outcross rate in the TOU-A population was estimated at 6% [82]. Local populations where killer and sensitive alleles coexist are therefore invaluable resources for further studies on PK3 dynamics in natural conditions. Moreover, the striking differences in haplotype frequencies between the populations reported in this study suggest that they were sampled at different steps of the dynamics. Further investigations of PK3 dynamics at the population level will provide important information for driving modeling and strategic studies, and nourish societal debate on the upcoming new types of plant drivers suggested for weed control or genetic rescue of endangered species [83,84].

The pollen killer PK3 in *A. thaliana* can be useful in understanding a novel mechanism of pollen abortion triggered in mitochondria. Moreover, it can become instrumental in documenting population dynamics of a segregation distorter in natural populations, enlightening theoretical and applied studies in evolutionary biology. Our results emphasize the value of the model *A. thaliana*, which initially was mainly a functional biology system, but is now also a powerful model in population biology relying on wealthy genetic resources associated with supporting molecular data. Multiple samplings within natural local populations together with constantly increasing *de novo* whole genome sequences will allow combining the advantages of molecular tools available in model species and of wild natural populations for deciphering the functioning and evolutionary dynamics of complex loci such as segregation distorters.

## Material and Methods

### Plant material and growth conditions

Natural accessions were provided by the Versailles Arabidopsis Stock Center (VASC) (https://publiclines.versailles.inrae.fr/). The ShaL3$^M$ and ShaL3$^H$ genotypes possess the Mr-0 cytoplasm and the Sha nuclear genome, excepted at the L3 locus, where they are homozygous Mr-0 and heterozygous Sha/Mr-0, respectively [50]. The early flowering version of the Mr-0 accession described in Simon et al [50] was used. The analysis of accessions from local populations was performed on plants grown from seeds collected in the wild. Plants were grown in the greenhouse under long-day conditions (16h day, 8h night) with additional artificial light (105 mE/m$^2$/s) when necessary.

### Monitoring pollen viability

Pollen viability was monitored from flower buds harvested just before anthesis. Anthers were dissected from at least two buds per genotype and observed under a light microscope after Alexander staining [85].

### CRISPR-Cas9 mutageneses

Optimal guide-RNAs (S16 Table) were designed thanks to the CRISPOR tool (http://crispor.gi.ucsc.edu/ [86]). For *APOK3-like*/*KPOK3C* mutagenesis, two different guides-RNAs were used, one (gRNA_APOK3-L#1) in the specific region of the gene, and the other (gRNA_APOK3-L#2) in the region identical to *APOK3*. The latter has also made it possible to obtain mutations in the gene *APOK3*. Guide-RNAs were synthetized by Twist Bioscience (https://www.twistbioscience.com/) under the control of the *A. thaliana* U6-26 RNA polIII promoter [87]. The resulting DNAs were cloned by the Gateway strategy in the binary vector pDe-CAS9DsRed expressing the CAS9 and the selectable marker DsRed [88]. The constructs were introduced into *Agrobacterium tumefaciens* strain C58C1 pMDC90, and then into the ShaL3$^M$ genotype by floral dipping. After the selection process, transformants were crossed with Sha

and hybrids that did not inherit the transgene were analyzed by PCR and sequencing for mutations in the target gene. Both *KPOK3C* and *APOK3* were sequenced in the mutants obtained with the gRNA_APOK3-L#2.

## Gene expression analyses

The 5'-end of the *KPOK3C* Mr-0 transcript was determined using the Invitrogen 5'-RACE system according to the manufacturer's instructions. A *KPOK3C* gene-specific cDNA was initiated with the primer Mr530B_R0 from 1.5 μg of leaf RNA treated with DNaseI. After purification on S.N.A.P. column, the cDNA was dC-tailed, then amplified with the primers AAP (Invitrogen) and Mr530B_R1. The PCR reaction was loaded on agarose gel, picked and re-amplified with the primer AUAP (Invitrogen) and the nested primer Mr530B_R2. The PCR product was sequenced with Mr530B_R3. Primers are displayed in S16 Table and S1 File.

Total RNAs were isolated from leaves and closed buds as described by Dean et al [89]. Reverse transcription was performed in 50 μL for 30 minutes at 50˚C using Maxima Reverse Transcriptase (Thermo Fisher), from 0.2 μg (for *KPOK3A*) or 1μg (for *KPOK3C*) RNA primed with oligodT. cDNAs were PCR amplified with the primers 460_F6/460_R6 for *KPOK3A* and Mr530B_F8/Mr530B_R5 for *KPOK3C* (S16 Table).

## Complementation experiments

DNA sequences of *KPOK3A* and *KPOK3C*, from respectively 1kb and 0.65kb upstream of the ATG start codon, to 0.5 kb downstream of the stop codon, were synthetized by Twist Bioscience (https://www.twistbioscience.com/) and cloned into the pUPD2 vector. Using the GoldenBraid strategy (https://gbcloning.upv.es [90]), each of these genes was combined with a transcriptional unit conferring red fluorescence to seeds (pCMV:DSRed:tnos [88]) into a pDGB3-omega2 binary plasmid. The *KPOK3C* sequence from the ATG start codon to 0.5 kb downstream of the stop codon was also amplified from the synthetized fragment and placed under the control of the pRPS5A promoter and upstream of the RbcS terminator and then associated with the pCMV:DSRed:tnos transcription unit using the GoldenBraid strategy. The resulting constructs were individually transformed into *Agrobacterium tumefaciens* strain C58C1 pMDC90 and integrated by floral dipping into appropriate genotypes. The transformants were selected on the red fluorescence of seeds, checked by PCR, and independent transformants were selected for their genotype at PK3 to analyze the bias in their progenies. Untransformed siblings were used as controls.

## Cellular localization of the killer proteins

The Mr-0 genomic sequences of *KPOK3A* and *KPOK3C* were amplified from the start codon to the end of the coding sequence, omitting the stop codon, using specific primers (S16 Table) and were inserted into a pUPD2 vector. A truncated version of *KPOK3C*, *ΔKPOK3C*, which lacks the 116 first codons was also introduced into pUPD2. Transcriptional units (TU) were constructed using the GoldenBraid strategy (https://gbcloning.upv.es/; [90]) as described in Simon et al [50]. Shortly, each TU associated the *UBQ10* promoter, the amplified sequence in fusion with the coding sequence of *mTurquoise2*, for *KPOK3A*, or *Citrine2*, for *KPOK3C* and *ΔKPOK3C*, and the *RbcS* terminator. Each new TU was associated into an omega2 binary vector, with the APOK3-RFP fusion [50] as a mitochondrial marker and the seed-specific fluorescent marker Fast-R (S5 Fig) as the transformation marker. The constructs were introduced into the ShaL3^M genotype by floral dipping. Cellular localization of fluorescent proteins was assessed on leaf epidermal cells of at least two independent transformants (T1) using a Leica SP8 confocal microscope. The sequential mode by lines was used to prevent leakage of

fluorescence between channels. RFP was excited at 561 nm and measured at 599–666 nm; mTurquoise2 was excited at 442 nm and measured at 452–525 nm; Citrine2 was excited at 514 nm and measured at 525–570 nm. Figure panels were build using EzFig (https://imagej.net/imagej-wiki-static/EzFig) in Fiji [91].

## Determination of the different forms of the three PK3 genes

All primers are shown in S16 Table. The form of *APOK3* was determined by Sanger sequencing with 530_R1 PCR products obtained with the primers 530_F1/530_R1. The form of *KPOK3A* was determined by two PCRs: one with the primers 460_F1/460_R7 differentiated the Mr-like form from a form common to the Sha-like and Col-like forms (indel in the 5' UTR), and the other with the marker 460InSha differentiated the Sha-like form from a form common to the Mr-like and Col-like forms (indel in the intron). The presence of *KPOK3C* was assessed by PCR with the primers Mr530B_F4/530_R1(S1 File) and the presence of an insertion in *AT3G62610* by PCR with the primers AT3G62610_F1/AT3G62610_R1(S1 File).

## Sequence analyses

The *KPOK3C* gene structure was predicted by submitting the Mr-0 genomic sequence inserted in the *AT3G62610* gene (10,960 bp) to Eugene online (https://bioinformatics.psb.ugent.be/webtools/EuGene/) [51]. The predicted protein sequences of KPOK3A and KPOK3C were searched for homologies using the blastp tool online (https://blast.ncbi.nlm.nih.gov/Blast.cgi; [92]) with standard parameters, which returned putative conserved domains [93]. Predictions of cellular localization of proteins were made online using TargetP 2.0 (https://services.healthtech.dtu.dk/services/TargetP-2.0/)[94] and MULocDeep (https://www.mu-loc.org/) [53].

## Sequence alignments

Sequences of the PK3 locus (from *AT3G62440* to *AT3G62630*), were extracted from genome assemblies of 131 accessions [54,55] using a custom Python script (v.3.9.0). For each gene of the pollen killer, homologous sequences were searched using the blastn program of Blast+ (version 2.15.0+). The query sequences used were (1) the genomic sequence of the Col-0 *APOK3* gene, from 89 bp upstream of the ATG start codon to the stop codon; (2) the genomic sequences of Sha and Mr-0 *KPOK3A* genes, from 89 and 100 bp upstream of the ATG to the stop codon, respectively (taking into account the 11 bp insertion in the 5'-UTR of the Mr-0 form) because the large insertion in the intron in the Sha-like forms of the gene precluded a correct identification of start and end positions of both forms when a unique query sequence was used; (3) the Mr-0 genomic sequence of *KPOK3C*, from the ATG start codon to the stop codon. We used 80 percent of query coverage per hsp, and maximum hsps were adjusted to 15 for *APOK3*, and to 4 for the other genes, after checking that no accession gave the maximum set number of hits. The sequences of all full-length genes were extracted from the genomic sequences using the bedtools (version 2.30.0) getfasta program and the start and stop positions from the blast output. For each PK3 gene, the retrieved sequences were aligned using the workflow FastTree/OneClick (bootstrap: 1000 replicates) of the ngphylogeny website (https://ngphylogeny.fr/workflows/advanced/ [95]). The resulting Mafft alignments were used to establish the tables of polymorphisms. The *APOK3* tree was drawn as an unrooted dendogram with normalized supports, since no outgroup was available. The *KPOK3A* tree was rooted by including in the alignment the genomic sequence of *A. suecica* as an outgroup (GenBank accession JAEFBJ010000010.1, locus_tag ISN44_As10g032330).

## Supporting information

**S1 File. Sequence of the Mr-0 insertion in *AT3G62610*.**
(DOCX)

**S1 Fig. Pollen viability of mutants.**
(PDF)

**S2 Fig. Validation of the *APOK3-like* (*KPOK3C*) gene structure predicted by EuGene.**
(PDF)

**S3 Fig. Expression of *KPOK3A and KPOK3C* in leaves of diverse genotypes.**
(PDF)

**S4 Fig. Unrooted phylogenetic tree of *APOK3* sequences.**
(PDF)

**S5 Fig. Use of the FAST-R seed-specific fluorescent marker for Arabidopsis transformant selection.**
(PDF)

**S1 Table. Segregations at the PK3 locus in selfed progenies of crosses between Sha and KO mutants of *AT3G62460* in ShaL3$^{M}$.**
(XLSX)

**S2 Table. Segregations at the PK3 locus in selfed progenies of crosses between Sha and KO mutants of *APOK3-like* (*KPOK3C*) in ShaL3$^{M}$.**
(XLSX)

**S3 Table. Segregations at the PK3 locus in selfed progenies of recombinants complemented with the Mr-0 genomic sequence of *KPOK3A*.**
(XLSX)

**S4 Table. Segregations at the PK3 locus in selfed progenies of crosses between Ct-1 and KO mutants *apok3-1* and/or *apok3-2* in ShaL3$^{M}$.**
(XLSX)

**S5 Table. Segregations at the PK3 locus in selfed progenies of crosses between Mr-0 and accessions with a TAC form of *APOK3*.**
(XLSX)

**S6 Table. Segregations at the PK3 locus in progenies of crosses of different natural accessions with Sha.**
(XLSX)

**S7 Table. Forms of each of the three PK3 genes in 728 accessions.**
(XLSX)

**S8 Table. *KPOK3A* polymorphisms.**
(XLSX)

**S9 Table. *KPOK3C* polymorphisms.**
(XLSX)

**S10 Table. *APOK3* polymorphisms.**
(XLSX)

**S11 Table. PK3 haplotypes of 301 TOU-A accessions.**
(XLSX)

**S12 Table. PK3 haplotypes found in 11 local populations from Burgundy.**
(XLSX)

**S13 Table. Segregations at the PK3 locus in selfed progenies of crosses between Sha and Mr-0, and between Sha and accessions from Burgundy with killer haplotypes.**
(XLSX)

**S14 Table. Segregations at the PK3 locus in selfed progenies of crosses between Mr-0 and accessions from Burgundy with sensitive haplotypes.**
(XLSX)

**S15 Table. Segregations at the PK3 locus in selfed progenies of crosses between killer and sensitive accessions from Burgundy.**
(XLSX)

**S16 Table. Oligonucleotides used in this study.**
(XLSX)

## Acknowledgments

We are grateful to Katia Belcram and Alice Vayssières for their advices on confocal microscopy experimentation, and figure assembly, to Nadia Bessoltane for her help with bioinformatic tools, and to Mathilde Grelon and Olivier Loudet for their comments during the manuscript preparation. We acknowledge the excellent work of Mathis Prod'homme, Camille Charpentier and Olivia Jean-Bart during their internship stays. We thank Qichao Lian for sharing the R script used to plot accessions on the world map. We are grateful to the Genotoul bioinformatics platform Toulouse Occitanie (Bioinfo Genotoul, https://doi.org/10.15454/1.5572369328961167E12) for providing computing and storage resources.

The IJPB benefits from the support of Saclay Plant Sciences-SPS (ANR-17-EUR-0007). This work has benefited from the support of IJPB's Plant Observatory platforms PO-Cyto for microscopy experiments, PO-Plants for plant growing, and PO-VASC for Arabidopsis resources.

## Author Contributions

**Conceptualization:** Anthony Ricou, Matthieu Simon, Françoise Budar, Christine Camilleri.

**Formal analysis:** Anthony Ricou, Françoise Budar, Christine Camilleri.

**Investigation:** Anthony Ricou, Rémi Duflos, Françoise Budar, Christine Camilleri.

**Resources:** Marianne Azzopardi, Fabrice Roux.

**Writing – original draft:** Françoise Budar, Christine Camilleri.

**Writing – review & editing:** Matthieu Simon, Fabrice Roux, Françoise Budar, Christine Camilleri.

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
