## [Decision Letter · Decision Letter 0]

15 Nov 2024

PGENETICS-D-24-01160Identification of novel genes responsible for a pollen killer present in local natural populations of Arabidopsis thalianaPLOS Genetics Dear Dr. Camilleri, Thank you for submitting your manuscript to PLOS Genetics. After careful consideration, we feel that it has merit but does not fully meet PLOS Genetics's publication criteria as it currently stands. Therefore, we invite you to submit a revised version of the manuscript that addresses the points raised during the review process. Please submit your revised manuscript within 30 days Dec 15 2024 11:59PM. If you will need more time than this to complete your revisions, please reply to this message or contact the journal office at plosgenetics@plos.org. Please include the following items when submitting your revised manuscript:*
A rebuttal letter that responds to each point raised by the editor and reviewer(s). You should upload this letter as a separate file labeled 'Response to Reviewers'. This file does not need to include responses to formatting updates and technical items listed in the 'Journal Requirements' section below.*
A marked-up copy of your manuscript that highlights changes made to the original version. You should upload this as a separate file labeled 'Revised Manuscript with Track Changes'.*
An unmarked version of your revised paper without tracked changes. You should upload this as a separate file labeled 'Manuscript'. If you would like to make changes to your financial disclosure, competing interests statement, or data availability statement, please make these updates within the submission form at the time of resubmission. Guidelines for resubmitting your figure files are available below the reviewer comments at the end of this letter. We look forward to receiving your revised manuscript. Kind regards, Tanja SlotteSection EditorPLOS Genetics Tanja SlotteSection EditorPLOS Genetics Aimée DudleyEditor-in-ChiefPLOS Genetics Anne GorielyEditor-in-ChiefPLOS Genetics **Additional Editor Comments (if provided):** Thank you for submitting your manuscript to PLoS Genetics! We have now received reviews of your manuscript from three experts in the field. All reviewers were very positive, commenting that the work was "strong", "engaging" and "exciting". That said, they also suggest minor changes to improve the manuscript and the readers' abilities to understand the results. Please address the reviewers' suggestions in a revised version of the manuscript.  **Journal Requirements:****Reviewers' comments:** Reviewer's Responses to Questions

**Comments to the Authors:**

Reviewer #1: Ricou et al use a combination of genetic analyses, cell biology and population analyses to significantly advance understanding of the PK3 poison-antidote killer meiotic driver, previously discovered in the model plant A. thaliana. Previous work had identified the antidote of the system, APOK3 and showed that two regions (A and C) flanking APOK3 are required for killing. This work demonstrated that one copy of APOK3 can be sufficient to provide the antidote function and identified a novel APOK3 allele that is weakly resistant to the driver’s poisons. This work also identifies two poison genes, KPOK3A and KPOK3C, that are each necessary for the pollen killing. The authors were also able to show that KPOK3A is the only gene required for the killing in the ‘A’ region. Similar experiments to test if KPOK3C was the only killer in the ‘C’ region failed, but were informative in they suggest specific control of KPOK3C expression is important. The authors broadly sampled A. thaliana isolates and identified several alleles of both killer genes. They also analyzed haplotype frequencies and characterized haplotype phenotypes via crosses. Finally, they provided phylogenetic analyses of the three genes to provide insight on their evolutionary histories. Finding killer and sensitive haplotypes within populations is especially exciting. This is a strong paper, and I enjoyed reading it. With one exception, noted below, the claims were sufficiently supported by the data. My suggestions for the authors to consider are listed below.

Major suggestion:

The abstract claims the data ‘indicates that the pollen killer activity involves the mitochondria.’ I suggest rewording that claim as the molecular mechanisms of the killing are still unclear. This could be as easy as saying the data ‘suggest’ rather than ‘indicates.’ I agree that mitochondrial localization and phenotypic changes depending on the cytoplasmic background of the hybrids are both consistent with mitochondrial function, but more is required to conclusively demonstrate it. I also think it is important to state: 1) if you tested the phenotype of the tagged genes to see if they remained active and 2) if you performed western blots to see if the fluorescent proteins were staying attached to full length proteins. In my own experience, I have had tags disrupt function and had tags be clipped from proteins and end up in mitochondria.

Minor suggestions/comments:

The readability of the paper could be significantly improved by adding more extensive labeling to the figures. I could always find the information I needed to fully understand the figures by reading the legends. Still, I found myself adding guiding texts and labels to several figures, so I did not have to keep reading the figure legend each time I returned to a figure. I will list the additions I made to my copy of the figures; in case the authors decide to add some of these things to help guide future readers.

-There are several distinct things illustrated in panel 2A. Perhaps break into more panels (e.g. one for KPOK3A and 3C). It is also unclear where the presented KPOK3C sequence is relative to the gene cartoon.

-Figure 2B and other such plots: More complete genotype labels on the X-axes and formal labels on the Y-axes (e.g. % homozygotes in F2 progeny).

-Figure 3D: The labels 1 and 2 could be replaced with more informative labels

-Figure 4, 8, 10 (the pinheads) and 11: A key for the color coding would be helpful.

-Figure 9: I think this got distorted in PDF production, but the colors are not very easy to distinguish.

Many mutations are described as knock-outs, but all seem to leave coding sequence intact, leaving open the door for remaining gene function.

Are the oligos used in 3B on spots that are invariant?

Line 143: were these gene introductions random, or targeted in some way?

More information in the text/figure legend about what tissue is shown in Figure 4 is required.

Line 217: I was confused by ‘PK3 locus against the Mr-0 allele’ as I was thinking of the wt Mr-0 allele. This could be avoided by replacing ‘Mr-0 allele’ with ‘double mutant allele.’

Line 255: I think the table S1 should be S6.

Table 3: It would be great to include the % sha homozygotes as a column in this table. The new data from this paper is in S6, but it would be nice to see it all together.

Line 380: The t-haplotype in mouse has more than one killer gene. I think those act additively, rather than co-dependently.

Line 515: Might those be due to non-allelic gene conversion, a major driver of WTF gene evolution?

The framing of your work in the introduction and discussion was very nice.

The use of red and black type within the some of the figures to distinguish killer and sensitive types was very helpful.

The supplemental data tables were nicely organized and easy to navigate.

Reviewer #2: The manuscript “Identification of Novel Genes Responsible for a Pollen Killer in Local Natural Populations of Arabidopsis thaliana” by Ricou et al. investigates pollen killer genes using both global accessions and local populations of A. thaliana. The manuscript presents and discusses results in a clear, logical, and well-written manner, with high-quality figures throughout. In this study, the authors identified killer components within the previously known pollen killer locus, PK3. Leveraging available sequence data for this killer element, they found evidence suggesting its association with mitochondria. They also confirmed that killer activity was pollen-specific and, through extensive sequence analysis, identified distinct haplotypes for the killer locus, enabling them to identify additional killer accessions. Interestingly, they discovered that killer and non-killer haplotypes co-exist in natural populations, raising questions about why these killer alleles persist in nature and whether they might confer benefits under certain backgrounds or conditions. This paper contains a substantial amount of well-presented data that was highly engaging to read.

In only have a few minor suggestions:

In the Introduction, lines 69-87 could be revised to clarify which results are previously established findings and which are novel contributions from this study. Additionally, referencing result figures in the Introduction may not be common practice and could be reconsidered.

Figure 2: For clarity in point B, it would help to label the control as F1 (or is it F2?). Also, please add a y-axis label to indicate what the percentage represents. The same applies to Fig. 3B, Fig. 5B, and Fig. 6; please add y-axis labels to these figures for consistency.

Figure 7B: Could you specify what "frequency" refers to in this figure? Additionally, it would be helpful to explain how the sum of frequencies could exceed 100%.

Lastly, regarding the wild populations, since no resistant types were observed, can it be ruled out that such resistance alleles might be deleterious or, if pollen-specific, would not lead to seed production? Was the analysis of local populations based on tissues collected directly from the wild, or were seeds collected from the wild and then grown in the greenhouse? This information is currently not specified in the Materials and Methods section

Reviewer #3: In their manuscript “Identification of novel genes responsible for a pollen killer present in local natural populations of Arabidopsis thaliana” the authors first identify 2 “killer” genes involved in a 3-gene poison-antidote system that naturally segregates in the model system Arabidopsis thaliana, and for which they have already identified the antidote. They find that these two genes border the antidote and are tightly genetically linked, as would be expected by theory. They then go on to describe the cellular localization/targets of the poison (and antidote), finding that they co-localize to the mitochondria. They then describe the global distribution of this system, finding incredible polymorphism and that both a the haplotype that lacks the antidote and the killer haplotype have likely descended from a COL-0 like allele that possess the antidote, but lacks the killer genes, providing evidence for an “antidote first” model. Lastly, they discover that killer/resistant/sensitive alleles segregate on fine spatial scales (i.e. within populations), making it a potentially active killer-antidote system in nature, and describe a range of killer-like and resistant-like behaviors in lines from the same population.

The manuscript is a tour de force, genetically speaking, and in my opinion has gone above and beyond to rigorously describe an exciting system which will interest both evolutionary biologists and molecular geneticists alike. I offer only a handful of very minor suggestions. Congrats to the authors!

Minor

- There are a few locations where Arabidopsis is not italicized throughout the manuscript, so should be double checked.

- Line 25: grammatically, I think the sentence should read either “…in local populations remains…” or “…in a local population remains…”

- Line 54: the word “favoring” suggests will/intention. I would replace with “resulting in” or similar.

- Line 270- the authors state that they cannot access the 1001 genomes because they have been aligned to Col-0. Is it not possible to access the original fastq files and align them to one of the newer de novo genomes from Lian et al. 2024. Nature Genetics in which this region is present?

- Line 537: I don’t think “coherent” is the correct word- “consistent”, perhaps?

- Figure 9 is very blurry and hard to read, and I recommend the authors use a higher image quality in the final version.

**Have all data underlying the figures and results presented in the manuscript been provided?**

Reviewer #1: Yes

Reviewer #2: Yes

Reviewer #3: Yes

PLOS authors have the option to publish the peer review history of their article (what does this mean?). If published, this will include your full peer review and any attached files.

Reviewer #1: No

Reviewer #2: No

Reviewer #3: No

 **Figure resubmission:** While revising your submission, please upload your figure files to the Preflight Analysis and Conversion Engine (PACE) digital diagnostic tool, https://pacev2.apexcovantage.com/. PACE helps ensure that figures meet PLOS requirements. To use PACE, you must first register as a user. Registration is free. Then, login and navigate to the UPLOAD tab, where you will find detailed instructions on how to use the tool. If you encounter any issues or have any questions when using PACE, please email PLOS at figures@plos.org. Please note that Supporting Information files do not need this step. If there are other versions of figure files still present in your submission file inventory at resubmission, please replace them with the PACE-processed versions. **Reproducibility:** To enhance the reproducibility of your results, we recommend that authors deposit laboratory protocols in protocols.io, where a protocol can be assigned its own identifier (DOI) such that it can be cited independently in the future. Additionally, PLOS ONE offers an option to publish peer-reviewed clinical study protocols. Read more information on sharing protocols at https://plos.org/protocols?utm_medium=editorial-email&utm_source=authorletters&utm_campaign=protocols

---

## [Decision Letter · Decision Letter 1]

19 Dec 2024

Dear Dr Camilleri,

We are pleased to inform you that your manuscript entitled "Identification of novel genes responsible for a pollen killer present in local natural populations of Arabidopsis thaliana" has been editorially accepted for publication in PLOS Genetics. Congratulations!

Yours sincerely,

Angela Hancock, Ph.D.

Academic Editor

PLOS Genetics

Tanja Slotte

Section Editor

PLOS Genetics

Aimée Dudley

Editor-in-Chief

PLOS Genetics

Anne Goriely

Editor-in-Chief

PLOS Genetics

Comments from the reviewers (if applicable):

The reviewers are satisfied with the revisions and all recommend acceptance. I agree and thank you for submitting to PLoS Genetics. I also agree with the reviewers that the research presented here is exciting and that this paper will represent an important contribution to the field.

Reviewer's Responses to Questions

**Comments to the Authors:**

Reviewer #1: The authors have addressed my concerns. The paper is an important contribution to the field.

Reviewer #2: I do not have any further comments or suggestions to the revised version of the manuscript. The authors have answered all queries improving the paper accordingly. This is really a beautiful work which was enjoyable to read.

Reviewer #3: I think the authors for their careful revisions. I really enjoyed the manuscript!

**Have all data underlying the figures and results presented in the manuscript been provided?**

Reviewer #1: Yes

Reviewer #2: Yes

Reviewer #3: Yes

PLOS authors have the option to publish the peer review history of their article (what does this mean?). If published, this will include your full peer review and any attached files.

Reviewer #1: No

Reviewer #2: No

Reviewer #3: No

**Data Deposition**

http://datadryad.org/submit?journalID=pgenetics&manu=PGENETICS-D-24-01160R1

**Press Queries**

---

## [Editor Report · Acceptance letter]

8 Jan 2025

PGENETICS-D-24-01160R1 

Identification of novel genes responsible for a pollen killer present in local natural populations of Arabidopsis thaliana 

Dear Dr Camilleri, 

We are pleased to inform you that your manuscript entitled "Identification of novel genes responsible for a pollen killer present in local natural populations of Arabidopsis thaliana" has been formally accepted for publication in PLOS Genetics! Your manuscript is now with our production department and you will be notified of the publication date in due course.

With kind regards,

Anita Estes

PLOS Genetics

On behalf of:
